# Balanced Active Inference

**Boyu Chen**[1]*, **Zhixiang Zhou**[1,2]*, **Liuhua Peng**[3]†, **Zhonglei Wang**[4]†

[1]School of Economics, Xiamen University, Xiamen, China
[2]Shanghai Innovation Institute, Shanghai, China
[3]School of Mathematics and Statistics, The University of Melbourne, Melbourne, Australia
[4]Wang Yanan Institute for Studies in Economics, Xiamen University, Xiamen, China
`boyuchen@stu.xmu.edu.cn, zhixiangzhou.stat@outlook.com,`
`liuhua.peng@unimelb.edu.au, wangzl@xmu.edu.cn`

## Abstract

Limited labeling budget severely impedes data-driven research, such as medical analysis, remote sensing and population census, and active inference is a solution to this problem. Prior works utilizing independent sampling have achieved improvements over uniform sampling, but its insufficient usage of available information undermines its statistical efficiency. In this paper, we propose balanced active inference, a novel algorithm that incorporates balancing constraints based on model uncertainty utilizing the cube method for label selection. Under regularity conditions, we establish its asymptotic properties and also prove that the statistical efficiency of the proposed algorithm is higher than its alternatives. Various numerical experiments, including regression and classification in both synthetic setups and real data analysis, demonstrate that the proposed algorithm outperforms its alternatives while guaranteeing nominal coverage. Our code is available at: `https://github.com/Uninfty/Balanced_Active_Inference`

## 1 Introduction

Machine learning has revolutionized data-driven fields, yet its success still hinges on access to high-quality labeled data, which is a critical component for reliable inference. This dependency on labeled data is particularly pronounced in precision-sensitive fields such as medical diagnostics [1], financial risk assessment [2], and remote sensing [3], where the accuracy of predictions directly affects decision making. However, labeling remains a costly and time-consuming process, resulting in a persistent gap between the abundance of unlabeled data and the limitation of annotated resources [4, 5]. Conventional methods, including random sampling and heuristic-based selection, lack systematic prioritization of informative instances, leading to inefficient labeling [6, 7].

Active learning addresses the labeling bottleneck by iteratively selecting uncertain instances to maximize label efficiency [7, 8, 9]. Extending it to statistical inference, active inference [10] strategically queries labels where the model exhibits high uncertainty using independent sampling, and makes statistical inference, including confidence intervals and hypothesis tests, based on the acquired labels. Combined with prediction-powered inference [11], integrating model predictions with limited labeled instances, active inference outperforms random sampling in terms of statistical efficiency. Nonetheless, its reliance on independent sampling induces variance inflation and often yields imbalanced datasets. These issues may degrade estimation performance [12], especially under systematic bias or distribution shifts, ultimately limiting the practical utility of existing methods.

---

*Equal contribution.
†Corresponding author.

Therefore, a critical challenge in active inference is to further improve statistical efficiency under a constrained labeling budget. To address the inefficiency introduced by traditional active inference strategies, we propose balanced active inference. This method strategically selects informative instances while maintaining statistical representativeness of the population. The key idea is to enforce structural balance in the selected samples so that they preserve key characteristics of the overall data distribution, thereby improving statistical efficiency without increasing the labeling budget. Our method builds upon the principle of covariate balancing, a classical technique in survey sampling and causal inference that aligns the distribution of sampled instances with population-level summaries of auxiliary variables [13, 14].

To enforce covariate balance in active instance selection, we implement balanced active inference using the cube method [15, 16], a sampling algorithm that by consecutively updating selection probabilities to satisfy certain balancing constraints. The cube method operates in two phases: a *flight phase*, which iteratively adjusts inclusion probabilities to approximate target distributions, and a *landing phase*, which finalizes the selection by resolving residual imbalances through constrained optimization.

Compared with existing works, the proposed balanced active inference framework offers three advantages. First, we reconceptualize model uncertainty estimates as dynamic auxiliary variables, enabling simultaneous optimization for informativeness and representativeness during instance selection. Second, we introduce a balancing condition that constrains the weighted sum of uncertainties among the labeled instances to match the corresponding population total, effectively preventing oversampling from specific uncertainty regions and promoting more stable estimates. Third, we provide a theoretical guarantee that he proposed balanced active inference framework yields lower asymptotic variance compared to conventional active inference methods.

Our contributions can be summarized as follows.

- Innovatively increase the statistical efficiency of active inference through balanced sampling.

- Incorporate model uncertainty as an auxiliary covariate in balanced sampling using a cube method.

- Provide closed-form expressions for the asymptotic variance, illustrating the variance reduction property of the proposed method.

- Demonstrate the broad applicability and superiority of our method through extensive experiments on diverse real-world and synthetic datasets.

**Related work.** *(1) Label Inference.* The challenge of drawing valid inferences from partially labeled data has inspired diverse methodological developments. Traditional methods for missing data, such as inverse probability weighting and multiple imputation [17, 18], established foundational principles for handling label scarcity. Semi-supervised inference methods [19, 20] demonstrated how unlabeled data could improve efficiency in parameter estimation, particularly under smoothness assumptions. A pivotal advancement emerged with prediction-powered inference (PPI) [11], which integrates machine learning predictions with a small labeled dataset to estimate population quantities. By treating predictions as noisy proxies for missing labels, PPI constructs debiased estimators while maintaining statistical validity. However, its reliance on uniform random sampling undermines its statistical efficiency, as it fails to prioritize informative instances for labeling.

*(2) Active Learning.* Active learning addresses label efficiency by adaptively selecting instances for annotation based on model uncertainty [7, 8, 9, 21]. While classical active learning focuses on optimizing model training [22], recent work extends these principles to statistical inference. Active inference [10] formalizes this paradigm, employing unequal-probability sampling to prioritize uncertain instances. By combining actively acquired labels with model predictions via a GD (general difference) estimator [23], it incorporates auxiliary information to correct for sampling bias and improve estimation efficiency. Despite its advantages, the independent sampling mechanism inherent to active inference introduces variance inflation, as independent label selections may yield imbalanced instances that poorly represent critical regions of the data distribution.

*(3) Balanced Sampling.* The use of auxiliary information in finite population sampling is widely recognized for enhancing estimation precision. Classical methods, such as stratification [12, 24] and probability proportional-to-size sampling [25, 26], exploit known auxiliary variables to reduce variance. More advanced balanced sampling techniques impose equality constraints, ensuring that

weighted sample summations match population totals, guaranteeing significant variance reduction [27, 28].

## 2  Problem setup

Suppose we have access to two datasets: a small labeled dataset $\mathcal{D}_l = \{(X_j, Y_j)\}_{j=1}^m$ with $m$ instances, independently and identically distributed (i.i.d.) from a distribution $\mathbb{P} = \mathbb{P}_X \times \mathbb{P}_{Y|X}$, and a large unlabeled dataset $\mathcal{D}_u = \{X_i\}_{i=1}^n$ with $n$ instances drawn i.i.d. from $\mathbb{P}_X$, where the corresponding labels $\{Y_i\}_{i=1}^n$ are unobserved. Denote $\mathcal{X} \subseteq \mathbb{R}^d$ as the feature space, and $\mathcal{Y} \subseteq \mathbb{R}$ as the label space. The primary goal is to estimate the population mean of the unobserved labels in $\mathcal{D}_u$, defined as

$$\theta^* = \mathbb{E}(Y_1). \tag{1}$$

To leverage feature-label relationships, a predictive model $\hat{f} : \mathcal{X} \to \mathcal{Y}$ is trained on $\mathcal{D}_l$. Additionally, a labeling budget of $n_b$ allows the query of labels for a subset of $\mathcal{D}_u$. Let $\xi_i \in \{0, 1\}$ denote an indicator for whether the label $Y_i$ is acquired for the instance $X_i \in \mathcal{D}_u$ with $\mathbb{E}[\sum_{i=1}^n \xi_i] = n_b$. The challenge lies in designing an estimator that optimally combines the predictive power of $\hat{f}(\cdot)$ with strategically sampled labels to reduce the variance of an estimator of (1).

Active inference employs a machine learning model $\hat{f}(\cdot)$ to predict labels for unobserved instances, coupled with an adaptive sampling strategy that corrects potential prediction biases. The sampling policy $\pi : \mathcal{X} \to [0, 1]$ determines label acquisition probabilities through uncertainty quantification. Specifically, let $\hat{u}(X_i)$ represent the model's estimated uncertainty measure for instance $i$, typically equal to $|Y_i - \hat{f}(X_i)|$. The sampling probabilities are then normalized as

$$\pi(X_i) = \frac{n_b}{n} \cdot \frac{\hat{u}(X_i)}{\frac{1}{n} \sum_{j=1}^n \hat{u}(X_j)}, \tag{2}$$

ensuring $\mathbb{E}(\sum_{i=1}^n \xi_i) = n_b$ through scaling. This allocation prioritizes regions where the model exhibits higher prediction uncertainty.

A GD estimator [23] of (1) for the inference,

$$\hat{\theta} = \frac{1}{n} \sum_{i=1}^n \left[ \hat{f}(X_i) + (Y_i - \hat{f}(X_i)) \frac{\xi_i}{\pi(X_i)} \right], \tag{3}$$

which is unbiased regardless of the form of the predictive model $\hat{f}(\cdot)$. Specifically, prediction-powered inference [11] emerges as a special case when $\pi(X_i) = n_b/n$, corresponding to uniform random sampling.

## 3  Balanced active inference

Our framework advances active inference by integrating balanced sampling through the cube method, which enforces structural constraints on auxiliary variables to improve statistical efficiency. Specifically, we impose the balancing condition

$$\sum_{i=1}^n \frac{\hat{u}(X_i)\xi_i}{\pi(X_i)} = \sum_{i=1}^n \hat{u}(X_i), \tag{4}$$

where $\hat{u}(X_i)$ quantifies the uncertainty of the predictive model $\hat{f}(X_i)$. The balancing constraint (4) ensures that the selected instances preserve the population structure of model uncertainties, mitigating selection bias. To operationalize this, we employ the cube method, a two-phase sampling algorithm that first iteratively adjusts inclusion probabilities to satisfy balancing constraints (flight phase), and then resolves residual imbalances via a landing phase when exact balancing is infeasible [16]; see Section S1 of Supplementary Material for an introduction to the cube method.

Once we obtain sampling indicators $\{\xi_i\}_{i=1}^n$ through the proposed balanced sampling strategy, we still consider a GD estimator

$$\tilde{\theta} = \frac{1}{n} \sum_{i=1}^n \left[ \hat{f}(X_i) + (Y_i - \hat{f}(X_i)) \frac{\xi_i}{\pi(X_i)} \right]. \tag{5}$$

Different from (3), the sampling indicators associated with (5) satisfy the balancing constraint (4).

**Remark 1.** *It may seem preferable to balance $\hat{f}(X_i)$ as well, since $\hat{f}(X_i)$ directly captures predictive information about $Y_i$. However, enforcing the balancing constraint on $\hat{f}(X_i)$ leads the estimator in (5) to degenerate into the form $n^{-1} \sum_{i=1}^{n}(Y_i \xi_i / \pi_i)$, thereby forfeiting any benefits from active sampling. In contrast, balancing on $\hat{u}(X_i)$ preserves the correction term in (5). If $\hat{u}(X_i)$ accurately approximates the residual error, i.e., $\hat{u}(X_i) \approx Y_i - \hat{f}(X_i)$, then the proposed estimator $\tilde{\theta} \approx n^{-1} \sum_{i=1}^{n} Y_i$, thereby improving the statistical efficiency of the GD estimator.*

**Uncertainty measures**  To effectively quantify prediction uncertainty for different task types, we define specific uncertainty measures tailored to regression and classification settings. For regression problems, the uncertainty is captured by the absolute residual $|Y_i - f(X_i; \hat{\theta})|$. For classification tasks, let $p(X_i) = (p_1(X_i), \ldots, p_K(X_i))$ represent the predicted class probabilities. The uncertainty measure is defined as $u(X_i) = \frac{K}{K-1} \left(1 - \max_{j \in [K]} p_j(X_i)\right)$, which attains its maximum when the model is maximally uncertain and decreases to zero when the model exhibits high confidence in a single class. The results of using other uncertainty quantification are provided in Section S6 in the Supplementary Material.

**Stabilization via mixed sampling**  As suggested by the traditional active inference literature [10], direct implementation of $\pi(X_i) \propto \hat{u}(X_i)$ risks instability when $\hat{u}(X_i)$ is misspecified. To safeguard against variance inflation from near-zero $\pi(X_i)$, we consider the following $\tau$-mixed rule:

$$\pi^{(\tau)}(X_i) = \tau \cdot \frac{n_b \hat{u}(X_i)}{\sum_{j=1}^{n} \hat{u}(X_j)} + (1 - \tau) \cdot \frac{n_b}{n}, \tag{6}$$

where $\tau \in [0, 1]$ controls the trade-off between uncertainty prioritization and robustness. Empirical analysis demonstrates that $\tau = 0.5$ achieves favorable bias-variance trade-offs across diverse scenarios, aligning with findings in [10]. This mixture ensures $\pi^{(\tau)}(X_i) > 0$ universally while retaining adaptivity to $\hat{u}(X_i)$. The relevant sensitivity analysis of $\tau$ is discussed in Section S5 in the Supplementary Material.

**Implementation**  Algorithm 1 outlines our cube-based balanced active inference procedure. The flight phase enforces the balancing constraint via iterative geometric projections, followed by a landing phase that minimizes deviation from target inclusion probabilities. The integration of the cube method with uncertainty-aware sampling distinguishes it from conventional balanced sampling, as the auxiliary variable $\hat{u}(x)$ directly links to the statistical efficiency of the GD estimator. The computational complexity of the cube method is $\mathcal{O}(n \times p^2)$ [16], where $n$ is the population size and $p$ the number of balancing covariates. In our implementation, the uncertainty measure $u$ is used as the only auxiliary covariate, and the complexity of balanced sampling reduces to $\mathcal{O}(n)$.

This synthesis of balanced sampling and active inference provides a unified framework for semi-supervised mean estimation, where model predictions guide sampling, while balancing constraints safeguard against distributional shifts—a key advancement over existing works.

**Extension to M-estimation**  Consider a general M-estimation problem that, given a class of functions $f(X_i; \theta)$,

$$\theta^* = \arg\min_{\theta} \mathbb{E}\left[L(X_1, Y_1; \theta)\right],$$

where $L(X_1, Y_1; \theta)$ is a loss function measuring the discrepancy between the true label $Y_1$ and the predicted label $f(X_1; \theta)$, and $\theta$ is the parameter of interest. The goal is to estimate $\theta^*$ using the labeled data $\mathcal{D}_l$ and the unlabeled data $\mathcal{D}_u$. Suppose that there have been an estimation $f(X_i; \hat{\theta})$ trained on the labeled data $\mathcal{D}_l$ and a uncertainty estimator $\hat{u}(X_i)$ trained on $L(X_i, Y_i; \hat{\theta}) - L(X_i, f(X_i; \hat{\theta}); \hat{\theta})$, a sampling scheme $\{\pi(X_i)\}_{i=1}^{n}$ is derived following (2) given a budget $n_b$. We use the cube method to generate an assignment $\{\xi_i\}_{i=1}^{n}$ such that the balancing constraint in (4) holds. The proposed estimator for M-estimation is then defined as

$$\tilde{\theta} = \arg\min_{\theta} \frac{1}{n} \sum_{i=1}^{n} \left\{ L(X_i, f(X_i; \hat{\theta}); \theta) + \left[ L(X_i, Y_i; \theta) - L(X_i, f(X_i; \hat{\theta}); \theta) \right] \frac{\xi_i}{\pi(X_i)} \right\}. \tag{7}$$

---
**Algorithm 1** Balanced active inference
---

1: Train a prediction model $\hat{f}(\cdot)$ on labeled training data $\mathcal{D}_l = \{(X_j, Y_j)\}_{j=1}^m$.
2: Compute residuals $e_j = \hat{f}(X_j) - Y_j$ and train an uncertainty model $\hat{u}(X_j) \approx |e_j|$.
3: Sample $n_b$ instances for labeling given the label budget $b = \mathbb{E}[n_b/n]$.
4: **for** $X_i \in \mathcal{D}_u$ **do**
5:     Predict label $\hat{Y}_i = \hat{f}(X_i)$.
6:     Predict uncertainty $\hat{u}_i = \hat{u}(X_i)$.
7:     Compute active probability $p_{a,i} = \frac{b \cdot \hat{u}_i}{\bar{u}}$, where $\bar{u} = \frac{1}{n}\sum \hat{u}_i$ and uniform probability $p_{e,i} = b$.
8:     Blend with uniform sampling: $\pi_i^{(\tau)} = \tau p_{a,i} + (1 - \tau)p_{e,i}$.
9: **end for**
10: Apply the cube method on $\{\pi_i^{(\tau)}\}_{i=1}^n$ to get $\{\xi_i\}_{i=1}^n$ satisfying the balancing constraint

$$\sum_{i=1}^n \frac{\hat{u}_i \xi_i}{\pi_i^{(\tau)}} = \sum_{i=1}^n \hat{u}_i.$$

11: Compute $\tilde{\theta}$ via (5).

---

The first term in the curly braces is the loss function evaluated at the predicted label $f(X_i; \hat{\theta})$, while the second term is the correction term that accounts for the difference between the loss function evaluated at the true label $Y_i$ and the predicted label $f(X_i; \hat{\theta})$. When the prediction $f(X_i; \hat{\theta})$ is poor, the correction helps to reduce the variance of the estimator by adaptively adjusting the inclusion probabilities based on the uncertainty of the predictions. By leveraging the cube method, we ensure that the selected instances are balanced with respect to the uncertainty estimates, leading to a more efficient estimation process.

## 4  Theoretical properties

Before presenting the main theoretical results, we introduce some generality assumptions under which our analysis is conducted.

**Assumption 1.** *There exists a function $f$, such that*

$$Y_i = f(X_i) + \varepsilon_i,$$

*where $\varepsilon_i$ satisfies $\mathbb{E}(\varepsilon_i | X_i) = 0$.*

**Assumption 2.** *Assume $\mathbb{E}(Y_1^2 + f(X_1)^2 + \hat{f}(X_1)^2 + \hat{u}(X_1)^2) < \infty$.*

**Assumption 3.** *There exists a positive constant $c \in (0, 1)$, such that*

$$\mathbb{P}\left(\xi_i = 1 \mid X_1, \ldots, X_n\right) \in [c, 1 - c].$$

**Assumption 4.** *For any $k \in \mathbb{N}$, we have with probability one,*

$$\lim_{n \to \infty} \sup_{i_1, \ldots, i_k} \left| \mathbb{E}\left(\prod_{j=1}^k \left(\xi_{i_j} - \pi_{i_j}\right) \mid X_1, \ldots, X_n\right) \right| = 0.$$

**Assumption 5.** *The estimator $\hat{f}(X_i)$ and $\hat{u}(X_i)$ statisfy*

$$\mathbb{E}\left\{ \left[ f(X_1) - \hat{f}(X_1) - \text{sgn}\left(f(X_1) - \hat{f}(X_1)\right) \hat{u}(X_1) \right]^2 \right\} = o(1),$$

*where $\text{sgn}(x) = I\{x > 0\} - I\{x < 0\}$ with $\text{sgn}(0) = 0$.*

Assumption 1 states that there exists an underlying regression function $f$ such that the observed outcomes $Y_i$ can be decomposed into a systematic component $f(X_i)$ and a zero-mean noise term $\varepsilon_i$, conditional on $X_i$. This is a standard assumption in supervised learning and nonparametric regression, ensuring the model is well-specified in the conditional expectation sense. Assumption 2 imposes a moment condition that ensures the second moments of the response variable $Y_i$, the true

regression function $f(X_i)$, and its estimator $\hat{f}(X_i)$ are all finite almost surely. Assumption 3 is standard in the semi-supervised inference literature and has been adopted in works such as [29]. Assumption 4 follows a conjecture from [30], asserting that as $n \to \infty$, the dependence among inclusion indicators for any fixed subset of instances vanishes asymptotically. Assumption 5 states that $\text{sgn}(f(X_i) - \hat{f}(X_i))\hat{u}(X_i)$ is a good estimator of $f(X_i) - \hat{f}(X_i)$, and we impose this to facilitate analytical tractability.

**Theorem 1** (Asymptotic normality for mean estimation). *Suppose Assumptions 1–5 hold,*

*(a) For the balanced active sampling scheme, the estimator $\tilde{\theta}$ defined in (5) satisfies*

$$\sqrt{n}(\tilde{\theta} - \theta^*) \xrightarrow{d} \mathcal{N}(0, V_0),$$

*where $V_0 = \mathbb{E}(\varepsilon_1^2/\pi_1) + \text{Var}[f(X_1)]$.*

*(b) For the traditional active inference with $\{\xi_i\}_{i=1}^n$ being independent, if $\mathbb{E}[\hat{f}(X_1)] = \theta^*$, the estimator $\hat{\theta}$ in (3) satisfies:*

$$\sqrt{n}(\hat{\theta} - \theta^*) \xrightarrow{d} \mathcal{N}\left(0, V_0 + \mathbb{E}\left[\left(f(X_1) - \hat{f}(X_1)\right)^2 \left(\frac{1}{\pi(X_1)} - 1\right)\right]\right).$$

The proof of Theorem 1 is provided in Appendix A. Theorem 1 establishes the asymptotic normality of the balanced active inference estimator. Specifically, the asymptotic variance of the proposed method consists of two components, including the scaled noise variance and the intrinsic variance of $f(X_1)$. Notably, this variance is reduced compared to that of the classical active inference methods, as the additional variability from estimation error in $\hat{f}(X_1)$ and non-uniform sampling is mitigated through the balancing constraint. Furthermore, the balanced active inference does not require the predictive model $\hat{f}(\cdot)$ to be unbiased, so it is more robust than existing active inference methods.

**Remark 2.** *Assumption 5 is introduced to enable the derivation of an explicit expression for the asymptotic variance of the proposed balanced active inference method, but it is not strictly necessary in practice. We conjecture that the proposed estimator retains its variance reduction benefits even when Assumption 5 is mildly violated. Empirical evidence in Section 5 supports our conjecture.*

## 5 Experiments

In this section, we evaluate the performance of our proposed method through both numerical simulations and real data applications. More numerical results about the real data analysis are shown in Section S3 of the Supplementary Material.

**Datasets** We consider three synthetic experiments, including a linear setup, a nonlinear setup and a Friedman setup [31]. Besides, we also consider real data applications, including six regression datasets and two classification datasets. Regression datasets include Bike Sharing [32], Communities and Crime [33], Concrete Compressive Strength [34], Energy Efficiency [35], Life Expectancy [36], Superconductivity Data [37], and binary classification datasets including Credit Fraud Detection [38] and Post-election Survey Research [39].

**Baselines** Our method (*cube-active*) is compared with three baseline methods across all experiments. As introduced in Section 2 and Section 3, the baselines include: (i) a simple random sampling labeling strategy using sample mean estimator $\hat{\theta} = \frac{1}{n_b} \sum_{j=1}^{n_b} Y_j$ with no involvement of machine learning models (*classical*); (ii) prediction-powered inference with GD estimator $\hat{\theta} = \frac{1}{n} \sum_{i=1}^n \left[ \hat{f}(X_i) + (Y_i - \hat{f}(X_i))\frac{\xi_i}{\pi} \right]$ using a uniform random labeling strategy (*uniform*); and (iii) active inference based on independent sampling strategies designed using machine learning model predictions with GD estimator (*traditional-active*).

**Evaluation metrics** For the four methods, we first report their Root Mean Squared Error (RMSE), which directly quantifies the deviation between point estimates and the true population mean. Additionally, leveraging the asymptotic normality of each method and its respective variance estimator, we compute the confidence intervals at a fixed confidence level 0.9, and compare their empirical coverage rates with respect to the true population mean. The variance estimators of each method are shown in Section S2 of the Supplementary Material.

**Experiment setup**   For all the datasets, the reported results are based on $T = 10\,000$ Monte Carlo simulations. Following recommendations from the traditional active inference literature [10], we set $\tau = 0.5$ in (6) across all experiments. All predictive models are obtained by XGBoost [40]. All experiments were conducted on a machine equipped with an Intel® Xeon® Gold 5118 CPU @ 2.30GHz, featuring 12 cores and 24 threads.

**Protocol**   The numerical analysis proceeds as follows. First, we generate $(X, Y)$ pairs and randomly split them into training/test sets. Then, an XGBoost regressor $\hat{f}(\cdot)$ is trained on the training set, and an uncertainty model $\hat{u}(\cdot)$ is fitted using XGBoost on $|\hat{f}(X_i) - Y_i|$ in $\mathcal{D}_l$. The specific model hyperparameters used for each dataset are detailed in Section S4 of the Supplementary Material. The details of computational resources and efficiency are provided in Section S3 of the Supplementary Material for completeness. The cube method is implemented by the R package `balancesampling` via Python's `rpy2` interface [41]. Other experimental details are provided in Section S4 in Supplementary Material.

## 5.1   Performance across diverse budgets

In this subsection, we evaluate the proposed method on three representative datasets under different labeling budgets, covering both synthetic and real-world scenarios. Specifically, we consider (i) a synthetic regression dataset generated by the nonlinear model

$$y = 10\sin(\pi x_1 x_2) + 20(x_3 - 0.5)^2 + 10x_4 + 5x_5 + \varepsilon,$$

where the predictors $x_1, \ldots, x_{10}$ are uniformly distributed on $[0, 1]$ and $\varepsilon$ follows a standard normal distribution; (ii) a real regression dataset (UCI Bike Sharing); and (iii) a real classification dataset (Credit Fraud Detection) with severe class imbalance. These datasets together provide a comprehensive test bed to evaluate the method's performance across different task types and data characteristics.

The results in Figure 1 reveal that the proposed cube-active method consistently outperforms alternatives across all labeling budgets. It achieves the lowest RMSE and the narrowest 90% confidence intervals, demonstrating superior predictive performance and sharper uncertainty quantification. Moreover, the empirical coverage rates closely match the nominal level in all cases, highlighting the validity and robustness of the inference procedure. Notably, even in the imbalanced classification setting, our method maintains stable coverage and tight intervals, highlighting its effectiveness across varying scenarios and labeling budgets.

## 5.2   Efficiency improvement

Table 1 presents the confidence interval widths (with empirical coverage rates in parentheses) for all methods under a labeling budget of $0.1$. Across the majority of datasets, the empirical coverage rates remain closely aligned with the nominal confidence level of $0.9$, confirming the validity of the asymptotic normal property and the statistical reliability of the proposed estimator.

Our proposed cube-active method achieves substantial and consistent efficiency gains over all baselines. Compared to traditional active inference based on independent sampling, cube-active reduces confidence interval widths by approximately $25\%$–$85\%$ across both synthetic and real-world datasets. When benchmarked against classical methods including uniform and simple random sampling, the improvement remains significant, with at least $30\%$ narrower intervals on all tasks.

These gains can be attributed to the enhanced covariate balancing induced by the cube sampling design, which effectively controls estimator variance by aligning the labeled set with the underlying feature distribution. As a result, cube-active achieves sharper inference and more precise uncertainty quantification without sacrificing coverage validity. This improvement is particularly valuable in low-budget regimes, where efficient use of labeled data is crucial for reliable statistical inference.

## 5.3   Label budget saving

Label budget saving refers to the percentage reduction in the required sample size by our method compared to raditional-active inference under a given estimation accuracy. We establish a precision benchmark using the confidence interval width of traditional-active inference at label budget $0.2$.

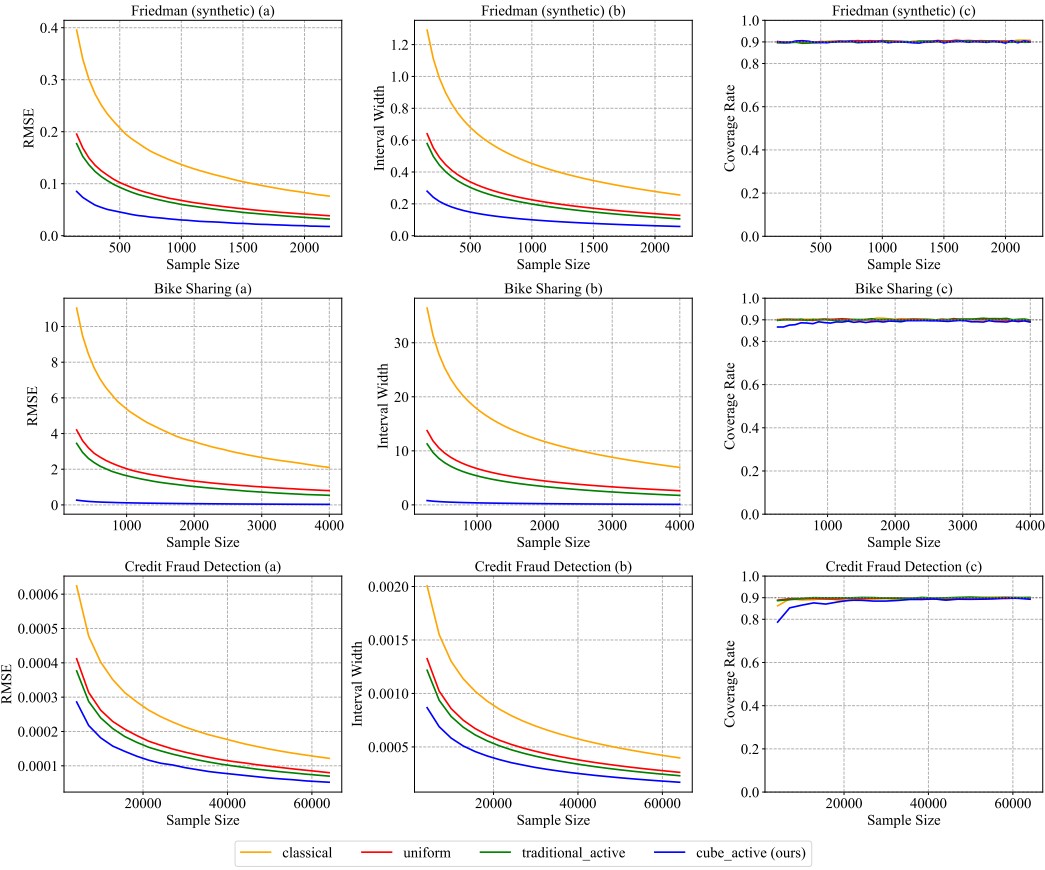

Figure 1: Performance comparison across three datasets with varying sample sizes. The top row (synthetic Friedman dataset), middle row (UCI Bike Sharing dataset), and bottom row (Credit Card Fraud Detection dataset) each display: (a) Root mean squared error (RMSE); (b) Average width of the 90% confidence intervals, reflecting inference precision; and (c) Empirical coverage rate, confirming interval estimation validity. The proposed cube-active sampling method consistently achieves superior performance across these datasets.

Table 1: Comparison of confidence interval width with 0.1 label budget across methods on 11 datasets. The bold values indicate the narrowest confidence intervals under valid coverage.

| Dataset | classical | uniform | traditional-active | cube-active |
|---|---|---|---|---|
| Linear (synthetic) | 0.3171 (0.8998) | 0.1690 (0.9053) | 0.1443 (0.8973) | **0.0722** (0.8998) |
| Nonlinear (synthetic) | 0.7786 (0.8931) | 0.3626 (0.8923) | 0.2963 (0.8967) | **0.1370** (0.9063) |
| Friedman (synthetic) | 0.6803 (0.8975) | 0.3798 (0.9040) | 0.3305 (0.8979) | **0.1052** (0.9056) |
| Bike | 19.2033 (0.9028) | 2.8213 (0.8987) | 2.2052 (0.8977) | **0.3316** (0.8969) |
| Communities | 0.0722 (0.8912) | 0.0516 (0.8942) | 0.0509 (0.8900) | **0.0466** (0.8885) |
| Concrete | 7.1217 (0.8970) | 3.9367 (0.9024) | 3.7415 (0.8999) | **2.6406** (0.8943) |
| Credit-fraud-detection | 0.0011 (0.8981) | 0.0007 (0.8930) | 0.0006 (0.8985) | **0.0005** (0.8700) |
| Energy | 5.1520 (0.8938) | 2.0364 (0.8954) | **1.8944** (0.9055) | 0.4136 (0.8025) |
| Life | 3.0828 (0.8967) | 1.4509 (0.8963) | 1.3050 (0.9036) | **0.7265** (0.8959) |
| Post-election | 0.0571 (0.8949) | 0.0426 (0.9019) | 0.0403 (0.8998) | **0.0381** (0.8980) |
| Superconductor | 3.2664 (0.9040) | 1.6711 (0.9067) | 1.5950 (0.9002) | **1.1680** (0.9003) |

For each dataset, we determine the minimal label budget required by alternative methods to achieve the most similar confidence interval width of the benchmark. Given our experimental grid of label budgets (0.03ˇ0.45 with 0.01 increments), we employ linear interpolation between adjacent budget points for precision matching. When a method's precision at the minimal tested budget (0.03) exceeds the benchmark, its required budget is conservatively denoted as "$< 10\%$". Conversely, if precision remains below benchmark at the maximal budget (0.45), the required budget is cautiously reported as "$> 150\%$".

Table 2 quantifies the label budget efficiency required by different methods to match the precision benchmark of traditional-active at 0.2 label budget. Our method achieves substantial budget savings across all datasets, requiring less label budget of the benchmark to attain equivalent precision. In several cases, our method attains superior precision with less than $40\%$ of the benchmark budget, demonstrating exceptional statistical efficiency. Compared to uniform sampling, our method only needs $5\%$ or less labeled instances to achieve higher precision in real-world applications. These savings stem from our method's optimal balanced sampling that simultaneously maximizes information gain and minimizes distributional discrepancy. The method's adaptive balancing mechanism proves particularly effective in high-dimensional settings where conventional active learning methods exhibit diminishing returns due to covariate mismatch. This systematic budget reduction, coupled with maintained statistical validity, establishes cube-active as a practical solution for label-constrained inference scenarios.

Table 2: Comparison of budget saving with the precision benchmarks across methods on 11 datasets.

| Dataset | classical | uniform | traditional-active | cube-active |
|---|---|---|---|---|
| Linear(synthetic) | >150% | 127% | 100% | 37% |
| Nonlinear(synthetic) | >150% | 117% | 100% | 70% |
| Friedman(synthetic) | >150% | 123% | 100% | 33% |
| Bike | >150% | 147% | 100% | 63% |
| Communities | 150% | 103% | 100% | 90% |
| Concrete | >150% | 110% | 100% | 67% |
| Credit-fraud-detection | >150% | 110% | 100% | 63% |
| Energy | >150% | 113% | 100% | <10% |
| Life | >150% | 123% | 100% | 43% |
| Post-election | >150% | 107% | 100% | 93% |
| Superconductor | >150% | 110% | 100% | 63% |

### 5.4 Performance of M-estimation problems

Figure 2 summarizes the statistical performance of our method in estimating linear regression coefficients for selected variables across the Linear and Bike Sharing datasets. Owing to the complexity of the underlying estimation procedures, deriving closed-form variance expressions for the estimators is intractable. Instead, we report the RMSE under different settings, which serves as a robust indicator of statistical efficiency and allows for clear performance comparisons across methods.

The results in Figure 2 consistently highlight the advantages of the proposed cube-active sampling strategy. For both the synthetic linear dataset and the real-world Bike Sharing dataset, cube-active sampling yields substantially lower RMSE values compared to all baseline methods, including uniform sampling, classical inference, and traditional active inference. These improvements illustrate the method's ability to provide more accurate coefficient estimation, thereby enhancing the overall statistical efficiency of the inference procedure.

## 6 Discussion

In this paper, we propose balanced active inference and demonstrate its superior statistical efficiency theoretically and numerically in various synthetic setups and real data analysis; see Section S8 and S9 of Supplementary Material for the limitations and societal impacts. A natural direction for future research is to explore the extension of this method into a sequential active inference framework.

Transitioning balanced active inference to a sequential setting introduces intriguing yet challenging methodological considerations. The principal challenge lies in the dynamic balancing requirement.

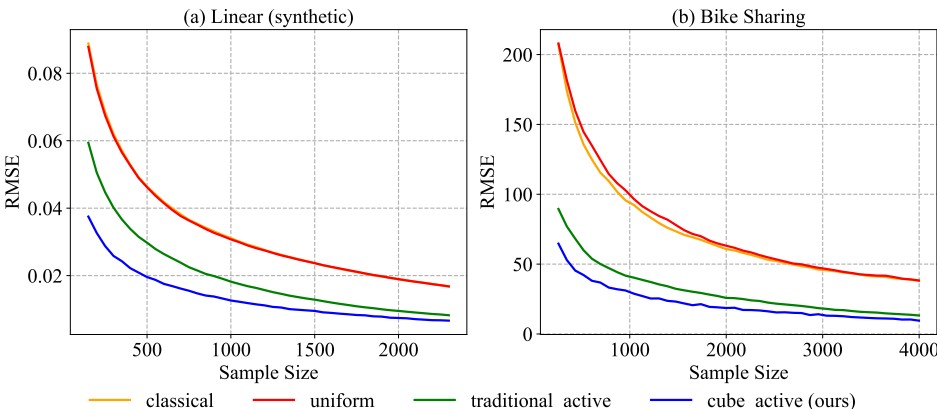

Figure 2: Root mean squared error (RMSE) of the least squares estimator in linear regression across two datasets with varying sample sizes. (a) the parameter of $x_1$ in synthetic Linear dataset ; (b) the parameter of *temperature* variable in Bike Sharing dataset. The proposed cube-active sampling method consistently achieves superior performance across these datasets.

In contrast to the batch scenario, where auxiliary variables are fixed, sequential balanced sampling involves continuously updating auxiliary variables when new instances become available. An adaptive implementation of the cube method, or a similar balanced sampling procedure, needs to integrate seamlessly with evolving model predictions and uncertainties in a computationally efficient manner. This dynamical adjustment could further reduce variance and improve representativeness, allowing for more precise allocation of labeling resources.

However, theoretical justification for such sequentially balanced sampling remains open. Extending existing results, such as martingale-based analyses that underpin sequential active inference, to balanced sampling contexts is not straightforward due to dependencies and complexity introduced by continually updated balancing constraints. Establishing rigorous statistical properties, such as unbiasedness, variance reduction, and asymptotic normality, within this sequential balanced framework will likely require novel analytical techniques or approximations.

Empirically, preliminary exploration through synthetic experiments and real data applications would be a valuable first step toward understanding sequential balanced active inference. Future studies should investigate various update strategies, examining how frequently and substantially the balancing conditions should be adjusted. Developing heuristics and computationally efficient algorithms capable of handling online updates of the balancing constraints would substantially advance the feasibility of sequential balanced active inference. Such advancements have great potential to further enhance labeling efficiency in practical applications, where data arrives continuously, and timely decision-making is critical.

## Acknowledgments and Disclosure of Funding

The authors would like to thank anonymous reviewers for their valuable comments. Wang's research is supported in part by Humanities and Social Sciences Foundation of the Ministry of Education of China Grant (No. 23YJA910005), NSFC (No. 12571291, 72533007, 71988101, 72033002). Peng was supported by ARC (Grant No. LP240100101).

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

# Appendix

## A Proof

In this section, we provide the detailed proof of Theorem 1. Before the formal proof, some technical lemmas are presented.

### A.1 Technical Lemmas

**Lemma 1** (Proposition 1 in [30]). *Suppose Assumptions 2 and 3 hold, and a cube method is conducted to balance auxiliary information $\{X_i : i = 1, \ldots, n\}$ under a sampling scheme $\{\pi_i : 1 = 1, \ldots, n\}$, then*

$$\frac{1}{n} \sum_{i=1}^{n} \frac{X_i \xi_i}{\pi_i} - \frac{1}{n} \sum_{i=1}^{n} X_i = o_p \left( \frac{q}{\sqrt{n}} \right),$$

*where $\xi_i \in \{0, 1\}$ indicates if the $i$-th instance is selected.*

**Lemma 2** (Lemma C.2 in [30]). *Let $f$ and $g$ be two functions such that for $f_i = f(\delta_i(1), \delta_i(0), X_i)$ and $g_i = g(\delta_i(1), \delta_i(0), X_i)$ we have $\mathbb{E}(f_i^2 + g_i^2) < \infty$ and $\mathbb{E}[f_i \mid X_i] = \mathbb{E}[g_i \mid X_i] = 0$.*

*If Assumptions 2, 3 and 4 hold. Then, conditional on $(X_i)_{i \geq 1}$,*

$$\frac{1}{\sqrt{n}} \sum_{i=1}^{n} f_i + g_i D_i \xrightarrow{d} \mathcal{N}(0, V_0)$$

*with $V_0 = \mathbb{E}\left[ f_1^2 + (2g_1 f_1 + g_1^2) \pi_1 \right]$.*

**Lemma 3** (Theorem 2 in [42]). *Let $U_n, V_n$ be two sequences of random variables and $\mathcal{B}_n$ be a $\sigma$-algebra. Assume that*
*(1) there exists $\sigma_{1n} > 0$ such that $\sigma_{1n}^{-1} V_n \to N(0, 1)$ in distribution as $n \to \infty$, and $V_n$ is $\mathcal{B}_n$ measurable;*
*(2) $E\{U_n \mid \mathcal{B}_n\} = 0$ and $\mathrm{Var}(U_n \mid \mathcal{B}_n) = \sigma_{2n}^2$ such that*

$$\sup_t \left| P\left( \sigma_{2n}^{-1} U_n \leq t \mid \mathcal{B}_n \right) - \Phi(t) \right| = o_p(1),$$

*where $\Phi(t)$ is the cumulative distribution function of the standard normal random variable;*
*(3) $\gamma_n^2 = \sigma_{1n}^2 / \sigma_{2n}^2 \to \gamma^2$ in probability as $n \to \infty$. Then*

$$\frac{U_n + V_n}{\sqrt{\sigma_{1n}^2 + \sigma_{2n}^2}} \to \mathcal{N}(0, 1)$$

*in distribution as $n \to \infty$.*

### A.2 Proof of Theorem 1

By definition, we have

$$\widetilde{\theta} - \theta^* = \frac{1}{n} \sum_{i=1}^{n} \left[ \hat{f}(X_i) + \left( Y_i - \hat{f}(X_i) \right) \frac{\xi_i}{\pi(X_i)} \right] - \mathbb{E}Y_1$$

$$= \frac{1}{n} \sum_{i=1}^{n} \left[ \hat{f}(X_i) - f(X_i) + \left( Y_i - \hat{f}(X_i) \right) \frac{\xi_i}{\pi(X_i)} \right] + \frac{1}{n} \sum_{i=1}^{n} \left( f(X_i) - \mathbb{E}Y \right). \tag{8}$$

For the first term in (8), we have

$$\frac{1}{n}\sum_{i=1}^{n}\left[\hat{f}(X_i) - f(X_i) + \left(Y_i - \hat{f}(X_i)\right)\frac{\xi_i}{\pi(X_i)}\right]$$

$$=\frac{1}{n}\sum_{i=1}^{n}\left[\hat{f}(X_i) - f(X_i) + \operatorname{sgn}\left(\hat{f}(X_i) - f(X_i)\right)\hat{u}(X_i)\right.$$

$$\left.+ \left(f_i - \hat{f}(X_i) + \varepsilon_i - \operatorname{sgn}\left(\hat{f}(X_i) - f(X_i)\right)\hat{u}(X_i)\right)\frac{\xi_i}{\pi(X_i)}\right]$$

$$-\frac{1}{n}\sum_{i=1}^{n}\hat{u}(X_i) + \frac{1}{n}\sum_{i=1}^{n}\hat{u}(X_i)\frac{\xi_i}{\pi(X_i)}$$

$$=\frac{1}{n}\sum_{i=1}^{n}\left\{\left[\hat{f}(X_i) - f(X_i) + \operatorname{sgn}\left(\hat{f}(X_i) - f(X_i)\right)\hat{u}(X_i)\right]\left(1 - \frac{\xi_i}{\pi(X_i)}\right)\right\} \qquad (9)$$

$$+\frac{1}{n}\sum_{i=1}^{n}\varepsilon_i\frac{\xi_i}{\pi(X_i)} + o_p\left(\frac{1}{\sqrt{n}}\right).$$

Denoting $\left[\hat{f}(X_i) - f(X_i) + \operatorname{sgn}\left(\hat{f}(X_i) - f(X_i)\right)\hat{u}(X_i)\right]\left(1 - \frac{\xi_i}{\pi(X_i)}\right)$ by $A_i$ and applying Chebyshev's inequality on (9), we have

$$\mathbb{P}\left(\left|\frac{1}{n}\sum_{i=1}^{n}A_i - \mathbb{E}A_i\right| \geq \frac{1}{\sqrt{n}}\right) \leq \frac{1}{n}\operatorname{Var}\left(\sum_{i=1}^{n}A_i\right). \qquad (10)$$

Further, we have $\operatorname{Var}\left(\sum_{i=1}^{n}A_i\right) = \sum_{i=1}^{n}\operatorname{Var}(A_i) + \sum_{i\neq j}\operatorname{Cov}(A_i, A_j)$. For $\operatorname{Var}(A_i)$, we have

$$\operatorname{Var}(A_i) = \mathbb{E}\left[\left(\hat{f}(X_i) - f(X_i) + \operatorname{sgn}\left(\hat{f}(X_i) - f(X_i)\right)\hat{u}(X_i)\right)^2\left(1 - \frac{\xi_i}{\pi(X_i)}\right)^2\right]$$

$$-\mathbb{E}\left\{\left[\hat{f}(X_i) - f(X_i) + \operatorname{sgn}\left(\hat{f}(X_i) - f(X_i)\right)\hat{u}(X_i)\right]\left(1 - \frac{\xi_i}{\pi(X_i)}\right)\right\}^2.$$

Since $\pi(X_i)$ is bounded from $c$ to $1 - c$, there exists a positive constant $C$ such that

$$\mathbb{E}\left[\left(\hat{f}(X_i) - f(X_i) + \operatorname{sgn}\left(\hat{f}(X_i) - f(X_i)\right)\hat{u}(X_i)\right)^2\left(1 - \frac{\xi_i}{\pi(X_i)}\right)^2\right]$$

$$\leq C \cdot \mathbb{E}\left[\left(\hat{f}(X_i) - f(X_i) + \operatorname{sgn}\left(\hat{f}(X_i) - f(X_i)\right)\hat{u}(X_i)\right)^2\right].$$

Further, by the assumption that $\mathbb{E}\left[f(X_i) - \hat{f}(X_i) - \operatorname{sgn}\left(\hat{f}(X_i) - f(X_i)\right)\hat{u}(X_i)\right]^2 = o_p(1)$, we have $\operatorname{Var}(A_i) = o_p(1)$. For $\operatorname{Cov}(A_i, A_j) = \mathbb{E}\left[A_i A_j\right] - \mathbb{E}\left[A_i\right]\mathbb{E}\left[A_j\right]$, first,

$$\mathbb{E}\left[A_i A_j\right]$$

$$=\mathbb{E}\left[\left(\hat{f}(X_i) - f(X_i) + \operatorname{sgn}\left(\hat{f}(X_i) - f(X_i)\right)\hat{u}(X_i)\right)\frac{\xi_i - \pi(X_i)}{\pi(X_i)}\right.$$

$$\left.\left(\hat{f}(X_j) - f(X_j) + \operatorname{sgn}\left(\hat{f}(X_i) - f(X_i)\right)\hat{u}(X_j)\right)\frac{\xi_j - \pi(X_j)}{\pi(X_j)}\right]$$

$$=\mathbb{E}\left\{\frac{\hat{f}(X_i) - f(X_i) + \operatorname{sgn}\left(\hat{f}(X_i) - f(X_i)\right)\hat{u}(X_i)}{\pi(X_i)}\right.$$

$$\cdot\frac{\hat{f}(X_j) - f(X_j) + \operatorname{sgn}\left(\hat{f}(X_i) - f(X_i)\right)\hat{u}(X_j)}{\pi(X_j)}$$

$$\left.\cdot \mathbb{E}\left[\left(\xi_i - \pi(X_i)\right)\left(\xi_j - \pi(X_j)\right)\Big|X_i, X_j\right]\right\}.$$

By Assumption 4, we have with probability 1, $\lim_{n\to\infty} \mathbb{E}\left[(\xi_i - \pi(X_i))(\xi_j - \pi(X_j)) \Big| X_i, X_j\right] = 0$. Thus, with probability 1, $\lim_{n\to\infty} \mathbb{E}\left[A_i A_j\right] = 0$. And by a similar argument, we have $\mathbb{E}\left[A_i\right] = \mathbb{E}\left[\left(\hat{f}(X_i) - f(X_i) + \mathrm{sgn}\left(\hat{f}(X_i) - f(X_i)\right)\hat{u}(X_i)\right)\frac{\xi_i - \pi(X_i)}{\pi(X_i)}\right] = 0$ as $n \to \infty$. Thus, it can be concluded that, with probability 1, $\lim_{n\to\infty}\sum_{i\neq j}\mathrm{Cov}(A_i, A_j) = 0$. Therefore, we have $\mathrm{Var}\left(\sum_{i=1}^n A_i\right) = o_p(1)$, then (10) implies that

$$\mathbb{P}\left(\left|\frac{1}{n}\sum_{i=1}^n A_i - \mathbb{E}A_i\right| \geq \frac{1}{\sqrt{n}}\right) = o_p(1).$$

Thus we have

$$\frac{1}{n}\sum_{i=1}^n\left\{\left[\hat{f}(X_i) - f(X_i) + \mathrm{sgn}\left(\hat{f}(X_i) - f(X_i)\right)\hat{u}(X_i)\right]\left(1 - \frac{\xi_i}{\pi(X_i)}\right)\right\} = o_p(\frac{1}{\sqrt{n}}).$$

By Lemma 2, we have conditional on $X_1, X_2, \ldots, X_n$

$$\frac{1}{\sqrt{n}}\sum_{i=1}^n \varepsilon_i \frac{\xi_i}{\pi(X_i)} \xrightarrow{d} \mathcal{N}\left(0, \mathbb{E}\left(\frac{\varepsilon_i^2}{\pi_i}\right)\right).$$

Thus, applying Lemma 3, we have

$$\sqrt{n}\left(\widetilde{Y} - \mathbb{E}Y\right) \xrightarrow{d} \mathcal{N}\left(0, \mathbb{E}\left(\frac{\varepsilon_1^2}{\pi_1}\right) + \mathrm{Var}\left[f(X_1)\right]\right).$$

If a Poisson sampling is used, by a standard CLT, we have

$$\sqrt{n}(\hat{\theta} - \theta^*) \xrightarrow{d} \mathcal{N}(0, V_1),$$

where

$$\begin{aligned}
V_1 &= \mathrm{Var}\left(\hat{f}(X_1) + \left(Y_1 - \hat{f}(X_1)\right)\frac{\xi_1}{\pi(X_1)}\right) \\
&= \mathrm{Var}\left[\mathbb{E}\left(\hat{f}(X_1) + \left(Y_1 - \hat{f}(X_1)\right)\frac{\xi_1}{\pi(X_1)}|X_1\right)\right] \\
&\quad + \mathbb{E}\left[\mathrm{Var}\left(\hat{f}(X_1) + \left(Y_1 - \hat{f}(X_1)\right)\frac{\xi_1}{\pi(X_1)}|X_1\right)\right] \\
&= \mathrm{Var}(Y_1) + \mathbb{E}\left[\left(Y_1 - \hat{f}(X_1)\right)^2 \frac{1 - \pi(X_1)}{\pi(X_1)}\right] \\
&= \mathrm{Var}(f_1) + \mathbb{E}\left(\frac{\varepsilon_1^2}{\pi_1}\right) + \mathbb{E}\left[\left(f(x) - \hat{f}(x)\right)^2 \left(\frac{1}{\pi(x)} - 1\right)\right].
\end{aligned}$$

Then, the proof is completed.

