# OpenReview forum: "Balanced Active Inference"
_NeurIPS.cc/2025/Conference — NeurIPS 2025 poster_

### Official Review · Reviewer_ibeC · 2025-07-02

**Clarity:** 2
**Significance:** 3
**Originality:** 3
**Rating:** 4
**Confidence:** 2

**Summary:**

This paper tackles the problem of active inference — estimating statistical properties of the labels in unlabeled datasets using a machine learning predictor. This is a challenging setting, as the predictor may introduce complex biases that must be carefully accounted for in the estimation process. The authors conduct a thorough theoretical analysis and propose a novel estimator that incorporates balancing constraints to achieve desirable properties, such as variance reduction and bounded asymptotic bias. These contributions are supported by both rigorous theoretical guarantees and empirical validation.

**Questions:**

Could you explain Remark 1?
Can you share the ablations that you performed to demonstrate that tau=0.5 is appropriate?

**Ethical Concerns:**

["NO or VERY MINOR ethics concerns only"]

**Final Justification:**

See my added comment below

**Limitations:**

Yes

**Quality:**

3

**Strengths And Weaknesses:**

This paper addresses a problem that I believe is important and relevant to the NeurIPS community. However, as someone who is not an expert in active inference, I found the paper somewhat difficult to follow. While I don’t feel equipped to assess the significance of the proposed method or the theoretical results in detail, I do expect NeurIPS papers to be accessible to an engaged, non-expert reader. In this case, while the mathematics was clear and well-presented it was difficult to distinguish between what aspects of the method were novel versus standard, which made it hard to assess the contributions. Some aspects and discussions were introduced very quickly (e.g. Remark 1 was not clear for me).

That said, I appreciated the framing of the theoretical results. While I cannot fully assess their depth or novelty, it was clear to me why they were relevant, and the assumptions were well-explained. As a non-expert, I focused more on the empirical evaluation (Section 5). I was concerned by the lack of ablations—for instance, the parameter τ=0.5 is used across all experiments with weak justification. Additionally, the choice of baselines seemed limited: many were discussed in the related work, but only three were used in the experiments. A justification for this selection would strengthen the empirical claims, though I defer to a more field-specific reviewer on this point. Overall, I thought the experimental setup was well-motivated, the repetitions and presentation were sound, and the results were clearly displayed and thoughtfully interpreted.

---

> ### Author Rebuttal · Authors · 2025-07-29
>
> # **Response to Reviewer ibeC**
>
> We sincerely thank you for recognizing the novelty of our methodological contribution, the thoroughness of our theoretical analysis, and the compelling empirical validation presented in our work, as well as for your thoughtful and constructive feedback, which is invaluable for refining the manuscript. Below, we address each of your concerns point by point:
>
> ## **Responding to Weaknesses**
>
> We appreciate your concern regarding the accessibility of our paper to non-experts in active inference. We acknowledge that while the mathematics in the paper is presented rigorously, some readers may find it difficult to distinguish the novel aspects of our method from existing techniques.
>
> > **What is novel and what is standard?**
>
> • **Standard components**: Active inference in the sense of Zrnić & Candès (2024) uses independent probability-proportional-to-uncertainty sampling together with an AIPW estimator.
>
> • **Our new contributions**:
> 1. We **reinterpret model uncertainty** as a continuous auxiliary variable and impose a population-level balancing constraint on it;
> 2. We are, to the best of our knowledge, the first to **integrate the cube method into active inference**.
> 3. We provide **closed-form asymptotic variance expressions** and prove that they are **strictly smaller** than those of independent sampling, yielding quantitative efficiency gains.
>
> We will revise the manuscript in the camera-ready version to make the contributions and methodological innovations more clearly distinguishable from standard components.
>
> ## **Responding to Questions**
>
> > ### **Q1. Explanation of Remark 1**
>
> We appreciate the reviewer’s comment regarding Remark 1. To clarify, Remark 1 states that if $\\hat{f}$ is balanced, we have the balance condition:
> $$ \\sum_{i=1}^n \\frac{\\hat{f}\\left(X_i\\right) \\xi_i}{\\pi\\left(X_i\\right)}\\approx\\sum_{i=1}^n \\hat{f}\\left(X_i\\right).$$
> Under this condition, the estimator simplifies as follows:
> \\begin{equation*}
>     \\begin{aligned}
>         \\tilde{\\theta}&=\\frac{1}{n} \\sum_{i=1}^n\\left[\\hat{f}\\left(X_i\\right)+\\left(Y_i-\\hat{f}\\left(X_i\\right)\\right) \\frac{\\xi_i}{\\pi\\left(X_i\\right)}\\right]\\\\
>         &=\\frac{1}{n}\\sum_{i=1}^nY_i\\frac{\\xi_i}{\\pi(X_i)} + \\frac{1}{n}\\left[\\sum_{i=1}^n \\hat{f}\\left(X_i\\right) - \\sum_{i=1}^n \\frac{\\hat{f}\\left(X_i\\right) \\xi_i}{\\pi\\left(X_i\\right)}\\right],
>     \\end{aligned}
> \\end{equation*}
> where the second term vanishes due to the balance condition. This shows that the estimator reduces to the standard inverse propensity weighted (IPW) estimator when $\\hat{f}$ is sufficiently balanced.
>
> > ### **Q2. Sensitivity Analysis on $\tau$**
>
> Thank you for this important suggestion. We acknowledge that the choice of $\tau = 0.5$ warrants explicit justification.
>
> - **Sensitivity analysis** on $\tau$ across multiple datasets will be added in the revised version.
> - The setup is: fix the sampling budget at 0.2 and compute the RMSE and interval width (analogous to Figure 1 in the main text, but with $\tau$ ranging from 0 to 1 on the x-axis) for methods.
> - Below is a representative subset of our experimental results:
>
> **Table Q2.1 Bike Sharing, RMSE**
>
> | $\\tau$ | `cube_active` | `poisson_active` | `uniform` | `lassical` |
> | ------- | ----------- | -------------- | ------- | --------- |
> | 0.00    | $\quad$ **0.127**   | $\quad$1.460          | 1.466   | 3.846     |
> | 0.25    | $\quad$**0.093**   | $\quad$1.261          | 1.466   | 3.846     |
> | 0.50    | $\quad$**0.079**   | $\quad$1.146          | 1.466   | 3.846     |
> | 0.75    | $\quad$**0.072**   | $\quad$1.064          | 1.466   | 3.846     |
> | 1.00    | $\quad$**0.130**   | $\quad$1.038          | 1.466   | 3.846     |
>
> **Table Q2.2 Bike Sharing,  Interval Width**
>
> | $\\tau$ | `cube_active` | `poisson_active` | `uniform` | `lassical` |
> | ------- | ----------- | -------------- | ------- | --------- |
> | 0.00    | $\quad$ **0.395**   | $\quad$ 4.847          | 4.844   | 12.805    |
> | 0.25    | $\quad$ **0.302**   | $\quad$ 4.128          | 4.844   | 12.805    |
> | 0.50    | $\quad$ **0.254**   | $\quad$ 3.754          | 4.844   | 12.805    |
> | 0.75    | $\quad$ **0.230**   | $\quad$ 3.530          | 4.844   | 12.805    |
> | 1.00    | $\quad$ **0.275**   | $\quad$ 3.432          | 4.844   | 12.805    |
>
> **Table Q2.3 Friedman, RMSE**
>
> | $\\tau$ | `cube_active` | `poisson_active` | `uniform` | `lassical` |
> | ------- | ----------- | -------------- | ------- | --------- |
> | 0.00    | $\quad$ **0.029**   | $\quad$ 0.069          | 0.068   | 0.137     |
> | 0.25    | $\quad$ **0.029**   | $\quad$ 0.063          | 0.068   | 0.137     |
> | 0.50    | $\quad$ **0.031**   | $\quad$ 0.060          | 0.068   | 0.137     |
> | 0.75    | $\quad$ **0.034**   | $\quad$ 0.060          | 0.068   | 0.137     |
> | 1.00    | $\quad$ **0.063**   | $\quad$ 0.079          | 0.068   | 0.137     |
>
> **Table Q2.4 Post-election survey research, RMSE**
>
> | $\\tau$ | `cube_active` | `poisson_active` | `uniform` | `lassical` |
> | ------- | ----------- | -------------- | ----------- | --------- |
> | 0.00    | $\quad$ **0.00841** | $\quad$ 0.00865        | 0.00862     | 0.01165   |
> | 0.25    | $\quad$ **0.00806** | $\quad$ 0.00840        | 0.00862     | 0.01165   |
> | 0.50    | $\quad$ **0.00790** | $\quad$ 0.00824        | 0.00862     | 0.01165   |
> | 0.75    | $\quad$ **0.00799** | $\quad$ 0.00839        | 0.00862     | 0.01165   |
> | 1.00    | $\quad$ 0.01014     | $\quad$ 0.01047        | **0.00862** | 0.01165   |
>
> **Table Q2.5 Credit Fraud Detection, RMSE**
>
> | $\\tau$ | `cube_active` | `poisson_active` | `uniform` | `lassical` |
> | ------- | ----------- | -------------- | ----------- | --------- |
> |    0.00 |   $\quad$ **0.000113** |   $\quad$    0.000142 | 0.000144 |  0.000219 |
> |    0.25 |   $\quad$ **0.000101** |   $\quad$    0.000132 | 0.000144 |  0.000219 |
> |    0.50 |  $\quad$  **0.000099** |   $\quad$    0.000129 | 0.000144 |  0.000219 |
> |    0.75 |   $\quad$ **0.000095** |   $\quad$    0.000128 | 0.000144 |  0.000219 |
> |    1.00 |  $\quad$  **0.000096** |   $\quad$    0.000128 | 0.000144 |  0.000219 |
>
> - All coverage rates in the above results closely approximate the target confidence level, confirming the validity of our findings.
> - Across all datasets, the RMSE and interval width **generally maintained the following ordering**: `cube_active < poisson_active < uniform < classical`.
> - At $\tau = 0$: The performance of `poisson_active` becomes nearly equivalent to `uniform`, as both rely solely on simple random sampling.
> - As $\tau$ increases, both `cube_active` and `poisson_active` exhibit **precision improvement** (decreasing RMSE and interval width), **reaching lower value** around $\tau = 0.5$ for different datasets. This demonstrably quantifies the precision gains enabled by active sampling.
> - When near $\tau = 1$, RMSE and interval width increase on some datasets. This occurs because the model for estimating prediction uncertainty ($\hat{u}$) is inherently imperfect. When the model erroneously assigns near-zero uncertainty ($\hat{u} \approx 0$) to specific data points, it can significantly inflate the estimator variance. These findings are well-aligned with conclusions established in Zrnic & Candès (2024).
>
> ## References
> - Zrnic, T., & Candès, E. J. (2024, July). Active statistical inference. In Proceedings of the 41st International Conference on Machine Learning (pp. 62993-63010).

---

> > ### Comment · Reviewer_ibeC · 2025-08-04
> >
> > Thanks for the comprehensive rebuttal that tackles my concerns about the experimental results and have rasied my score. However, as a non expert on this topic I am unconmfortable rasing my score further as I still worry about the overall readability of the paper --- this is very hard to judge without seeing the final version.

---

> > > ### Author Response · Authors · 2025-08-06
> > >
> > > Thank you for your response and for raising your evaluation score.
> > >
> > > As we revise our manuscript, we plan to:
> > > 1. **Introduction**
> > > - Systematically introduce the contributions and limitations of existing Active Learning/Inference research
> > > - Clearly articulate the problem our paper aims to address, namely reducing sampling-induced variance by balanced sampling
> > > - Highlight the **innovations** and **distinct contributions** of our approach from existing methods
> > > 2. **Problem setup & Balanced active inference**
> > > - Provide a **detailed overview** of existing Active Learning/Inference methods
> > > - Thoroughly describe the Cube Method
> > > - Clearly present our proposed improvements
> > > - Intuitively explain the **benefits** and **motivations** of our method
> > > 3. **Theoretical properties**
> > > - Provide a detailed explanation of our theoretical results
> > > - Offer more **intuitive interpretations** of the remarks and theorems (e.g., Remark 1)
> > > 4. **Experiments**
> > > - Evaluate the proposed method under a wider range of parameter settings
> > >
> > > We would be very grateful for any additional feedback. Thank you again for your thoughtful and constructive comments.

---

> > ### Comment · Reviewer_ibeC · 2025-08-07
> >
> > Thanks for the list of contributions. This would be great to see at the top of the final version of the paper. The explanation of remark 1 is helpful for my understanding and the additional sensitivity analysis for tau is very convincing, alongside its interpretation alongside the relevant theoretical results (although perhaps cube_active for Bike sharing RMSE shouldn't be emboldened for tau=0.130).

---

> > > ### Author Response · Authors · 2025-08-08
> > >
> > > We highly appreciate your additional comments. We have placed the list of contributions at the
> > > beginning and corrected the boldface issue for `cube_active` .

---

> ### Comment · Reviewer_ibeC · 2025-08-07
>
> Thanks for this. I believe that this is a great plan for improving clarity. Although it is very hard to increase score without seeing the resulting manuscript.

---

### Official Review · Reviewer_oZYC · 2025-07-02

**Clarity:** 3
**Significance:** 3
**Originality:** 3
**Rating:** 5
**Confidence:** 4

**Summary:**

This paper proposes a novel method for maximizing the limited labeling budget, called "balanced active inference", where balancing constraints are imposed based on the model uncertainty to guide the labeling process.
The goal is to achieve balanced sampling that effectively balances the trade-off between labeling cost and statistical efficiency.

**Questions:**

Please see and respond to questions in "Strenghts and Weaknesses".

**Ethical Concerns:**

["NO or VERY MINOR ethics concerns only"]

**Final Justification:**

The evaluation score has been increased after reviewing the authors' rebuttal to the reviewer's initial feedback.

**Limitations:**

The authors provide adequate discussions of the limitations of the current work as well as its potential societal impact in the supplemental material (sections S4 and S5).

**Quality:**

3

**Strengths And Weaknesses:**

1. The paper tackles an important problem that is of broad interest across diverse fields.
As labeling costs tend to be costly in many real-world applications, it is of significant practical and theoretical interest to devise an effective labeling strategy that can minimize the acquisition cost, maximize the efficacy of the acquired (additional) labels, and also ensure statistical efficiency (e.g., keeping the dataset "balanced").

2. Overall, the proposed approach is well-motivated and the paper proposes a reasonable solution by leveraging augmented
inverse propensity weighting (AIPW) estimator and the cube method for balanced sampling.

However, the paper would benefit from including a more in-depth discussion on relevant recent advances in active learning, prediction-powered inference, and balanced sampling to provide a clear context to the reader against which its novelty and significance can be judged.
For now, the discussion in Sec. 1 "Related work" is too short and not comprehensive enough.
The main novel and significant contributions made in this paper should be better outlined based on this expanded discussion of relevant work in the field.


3. While the paper provides some evidence that supports the potential benefits of the proposed balanced inference scheme, the current evaluation results are insufficient to properly assess its advantages (and any trade-offs made in order to keep the dataset balanced).

Especially:
(i) There is no comparison against widely used active learning strategies. It is important to show the pros and cons of "balanced active inference" compared to (uncertainty-based) AL strategies, and thereby show the impact of balanced sampling using the proposed balanced active inference scheme.
(ii) The proposed scheme utilizes an uncertainty model using XGBoost that tries to fit the prediction error. It is important to understand the impact of adopting different "uncertainty models". Especially, instead of predicting the error and using it as the uncertainty estimate, what would be the impact of utilizing different UQ (uncertainty quantification) schemes widely utilized in the field?
(iii) How does the balanced active inference scheme (differently) perform in easy vs. hard regression/classification problems?

4. For the classification problem, the authors utilize logistic regression.
As a result, the uncertainty model defined in the "Protocol" can be easily adopted without any change.
However, how would one estimate the uncertainty for different types of classifiers?
It would be helpful to discuss how the method can generalize.

5. Figure 1 lacks legends for the curves, which makes it difficult to interpret the results.

6. To provide a more comprehensive assessment of the reliability of interval estimation using the proposed technique, the authors should include coverage rates for different confidence levels -- in addition to what is considered in the current study (i.e., 0.9).

---

> ### Author Rebuttal · Authors · 2025-07-29
>
> # **Response to Reviewer oZYC**
>
> We sincerely appreciate your recognition of the problem's significance, the practical relevance of our work, and the soundness of our methodological approach. We also find your constructive feedback and critique extremely valuable for optimizing our manuscript. Below, we provide point-by-point responses to your specific suggestions:
>
> ## **Responding to Weaknesses**
>
> > ### **Weakness 3 (i). Active Learning Strategies in Inference**
>
> Thanks for your constructive feedback.
>
> - Our proposed "balanced active inference" method belongs to the emerging field of active statistical inference , which is fundamentally different from traditional active learning. Consequently, we respectfully argue that a direct comparison to AL strategies would be inappropriate.
>
> - The primary goal of traditional AL is to improve a **model's predictive performance** (e.g., accuracy, AUC) with minimal labeling cost (Deng et al., 2023).
>
> - In contrast, the goal of active inference is to improve the **efficiency** of **statistical inference**, i.e., to obtain narrower confidence intervals and more powerful p-values for population parameters, not to improve the model itself. Our work contributes to the latter.
>
> - AL query strategies typically select samples to maximize the model's information gain to better define its decision boundary.
>
> - Active inference sampling strategies are designed to optimally reduce the **asymptotic variance** of the statistical estimator. As formally derived by Zrnic & Candès (2024), this is a fundamentally different optimization problem tailored for inference, not learning.
>
> - Critically, it has been shown that naively applying AL for statistical estimation introduces significant sampling bias, which invalidates classical statistical procedures (Zhang et al., 2021).
>
> - In summary, comparing our inference-focused method against a learning-focused AL strategy would not be meaningful due to conflicting goals, methodologies, and evaluation metrics. The contribution of our work lies in achieving more efficient and robust statistical inference, and we believe it should be evaluated by those standards.
>
> > ### **Weakness 3 (ii) (iii). Discussion of Uncertainty Quantifications**
>
> Thank you for this insightful comment.
>
> - The proposed method belongs to the error-prediction paradigm in active inference. This strategy was established by Yoo & Kweon (2019) and continues to be an active and promising research direction, with recent work such as Sinha & Dongaonkar (2024).
>
> - We specifically chose to predict the **absolute error** (an L1-loss objective) due to its **robustness**. Unlike the L2-loss (mean squared error), the L1-loss makes the resulting uncertainty estimation **less sensitive to outlier** predictions. As noted in Hastie et al. (2009), the robustness of L1-norm-based methods against outliers is a fundamental principle in statistics and machine learning.
>
> - This approach is substantially more efficient than many widely-used UQ schemes. Methods like Deep Ensembles (Lakshminarayanan et al., 2017) require training multiple models, while MC Dropout (Gal & Ghahramani, 2016) necessitates numerous forward passes per sample. Our method requires only a **single** forward pass through a lightweight prediction head, making it highly **scalable and practical**.
>
> > ### **Weakness 4. Classification Problem**
>
> Thanks for raising concerns about classification problem. We formally extend the Balanced Active Inference framework to the general M-estimation setting, which cover the classification problem.
>
> #### **(i) Settings of M-estimation**
>
> * Given a family of functions $f(X_i;\theta)$, the task is to solve $ \theta^{\ast}= \arg\min_{\theta} \mathbb{E} \left [L\bigl(X_1, Y_1; \theta\bigr)\right]$.
> * $L(X_1, Y_1; \theta)$ measures the discrepancy between the true label $Y_1$ and the prediction $f(X_1;\theta)$.
> * Our goal is to estimate $\theta^{\ast}$ using a small labeled set $ \mathcal{D}_l$ and a larger unlabeled set $ \mathcal{D}_u $.
> * $ f(X_i;\hat{\theta}) $ is an initial estimator trained on $ \mathcal{D}_l $.
>
> #### **(ii) Uncertainty Measures**
>
> - **Classification.** Let $ p(X_i) = (p_1(X_i),\ldots,p_K(X_i))$ denote the predicted class probabilities. We define $$u(X_i)=\frac{K}{K-1}\Bigl(1-\max_{j\in[K]} p_j(X_i)\Bigr),$$
> which peaks when the model is maximally uncertain (uniform distribution) and drops to zero when the model is highly confident in one class.
>
> #### **(iii) M-Estimation**
>
> * A model $\hat{u}(\cdot)$ is trained on $\mathcal{D}_l$ to predict the uncertainty of points in  $\mathcal{D}_u $.
>
> * Given a labeling budget $n_b$, adopt the sampling scheme described in Equation (2) of the paper.
>
> * Generate an assignment via the cube method, ensuring the balancing constraint (Equation (4)) is satisfied.
>
> * The proposed estimator is
>
>   $$\tilde{\theta}=\underset{\theta}{\arg \min } \frac{1}{n} \sum_{i=1}^n\{L(X_i, f(X_i ; \hat{\theta}) ; \theta)+[L(X_i, Y_i ; \theta)-L(X_i, f(X_i ; \hat{\theta}) ; \theta)] \frac{\xi_i}{\\pi(X_i)}\},$$
>
>   where $\pi(X_i)$ is the inclusion probability.
>
> > ### **Weakness 5. Figure 1**
>
> Thanks for pointing out this issue.
> - The method `traditional_active` (referring to independent random sampling, also known as Poisson sampling) was inadvertently labeled as `poisson_active` in the original figure.
> - We will revise the figure accordingly.
>
> > ### **Weakness 6. Coverage Rates for Different Confidence Levels**
> - We have calculated the coverage rate under different confidence levels on multiple datasets.
> - Below are some experimental results with a fixed sampling budget of 0.35:
>
> |     $\quad$  $\quad$ $\quad$ $\quad$     Bike             |  Sharing         |     $\quad$  $\quad$        Credit Fraud            |  Detection |
> |-------------------------|----------------------|-------------------------|------------------------|
> | Confidence Level        | Coverage Rate        | Confidence Level        | Coverage Rate         |
> | 0.95                    | 0.9447              | 0.95                    | 0.9389                |
> | 0.99                    | 0.9889              | 0.99                    | 0.9836                |
>
> - The results demonstrate that our method is **close** to the preset confidence level in all cases.
> - The results will be included in the Supplement in the revised version.
>
> ## References
>
> - Deng, Y., Yuan, Y., Fu, H., & Qu, A. (2023). Query-augmented active metric learning. Journal of the American Statistical Association, 118(543), 1862-1875.
>
> - Zrnic, T., & Candès, E. J. (2024, July). Active statistical inference. In Proceedings of the 41st International Conference on Machine Learning (pp. 62993-63010).
>
> - Farquhar, S., Gal, Y., & Rainforth, T. (2021). On Statistical Bias In Active Learning: How and When to Fix It. In International Conference on Learning Representations.
>
> - Yoo, D., & Kweon, I. S. (2019). Learning loss for active learning. In Proceedings of the IEEE/CVF conference on computer vision and pattern recognition (pp. 93-102).
>
> - Haimovich, D., Karamshuk, D., Linder, F., Tax, N., & Vojnovic, M. (2024). On the convergence of loss and uncertainty-based active learning algorithms. Advances in Neural Information Processing Systems, 37, 122770-122810.
>
> - Hastie, T., Tibshirani, R., Friedman, J. H., & Friedman, J. H. (2009). The elements of statistical learning: data mining, inference, and prediction (Vol. 2, pp. 1-758). New York: springer.
>
> - Lakshminarayanan, B., Pritzel, A., & Blundell, C. (2017). Simple and scalable predictive uncertainty estimation using deep ensembles. Advances in neural information processing systems, 30.
>
> - Gal, Y., & Ghahramani, Z. (2016, June). Dropout as a Bayesian approximation: representing model uncertainty in deep learning. In Proceedings of the 33rd International Conference on International Conference on Machine Learning-Volume 48 (pp. 1050-1059).

---

> > ### Comment · Reviewer_oZYC · 2025-08-03
> > **Re: Rebuttal by Authors**
> >
> > I would like to thank the authors for providing a detailed response to my review comments.
> > The provided rebuttal has been helpful.
> >
> > Regarding the author's points w.r.t. comment 3 (ii), I understand the authors' points, which I find reasonable.
> > Having said that, I believe it would still be valuable to compare the proposed scheme with other methods (or variants of the proposed scheme that incorporate UQ estimates).
> > Despite the increased computational cost, can one gain by leveraging various UQ techniques (instead of solely focusing on prediction errors)?
> > Such comparison would provide additional insights into the proposed "balanced active inference" scheme and potential future directions.
> >
> > The authors have not provided a direct answer to the questions regarding "How does the balanced active inference scheme (differently) perform in easy vs. hard regression/classification problems?"
> > I think such a comparison would also provide valuable insights into the performance of balanced active inference scheme, proposed in this paper.
> >
> > Finally, regarding the authors' response to the reviewer's suggestion to compare their method with other (uncertainty-based) active learning schemes, I respectfully disagree.
> > While I recognize that "active inference" has a different objective from "active learning" and acknowledge their critical difference, it would still be valuable for readers to see how this change of "objective" in selecting future data points affects these different objectives: i.e., improved learning vs. improved statistical inference.
> > Understanding what one may potentially gain (or lose) would be informative, allowing them to choose between active learning vs. active inference schemes based on their respective goals/needs.
> >
> > In summary, I still have a few remaining concerns, which I hope the authors will be able to address.
> > Nevertheless, I am raising my original rating after reviewing the authors' rebuttal, which has addressed some of my initial concerns, and in consideration of the potential value of investigating the active inference problem.
> >
> > Thank you.

---

> > > ### Author Response · Authors · 2025-08-06
> > >
> > > Thank you for your response and for raising your evaluation score. Below, we address your specific points in details.
> > >
> > > > ## Uncertainty Quantification (UQ)
> > >
> > >  We have conducted additional experiments to evaluate the effectiveness of different uncertainty quantification (UQ) methods.
> > >
> > > - **Table 1.1**: results using **ensemble variance** as measure of uncertainty on the **Bike Sharing dataset**.
> > > - **Table 1.2**: results using **prediction interval** (5%–95% quantiles) from **quantile regression** as measure of uncertainty on the **Nonlinear dataset**.
> > > - Both **ensemble variance** and **prediction interval** show improved performance compared to baseline.
> > > - But both methods are inferior to **absolute error**.
> > >
> > >  **Table 1.1: Ensemble Variance (Bike Sharing, RMSE)**
> > >
> > > | UQ | `cube_active` | `poisson_active` | `uniform` | `classical` |
> > > | --- | --- | --- | --- | --- |
> > > | Absolute Error | **0.130284** | 1.776984 | 2.202898 | 5.792181 |
> > > | Ensemble Variance | 1.887203 | 1.992315 | 2.202898 | 5.792181 |
> > >
> > > **Table 1.2: Quantile Regression Evaluation (Nonlinear, RMSE)**
> > >
> > > | UQ | `cube_active` | `poisson_active` | `uniform` | `classical` |
> > > | --- | --- | --- | --- | --- |
> > > | Absolute Error | **0.080199** | 0.106123 | 0.117678 | 0.238968 |
> > > | Quantile Regression | 0.110491 | 0.112347 | 0.117678 | 0.238968 |
> > >
> > > > ## Evaluation on Easy vs. Hard Problems
> > >
> > > We conducted further experiments comparing easy and hard problem settings.
> > >
> > > - **Table 2.1**: Bike Sharing Dataset
> > >   - **Easy Problem**: uses all 13 features
> > >   - **Hard Problem**: uses only 3 features (temp, windspeed, registered)
> > > - **Table 2.2**: Credit Fraud Detection Dataset
> > >   - **Easy Problem**: uses all 30 features
> > >   - **Hard Problem**: uses 4 features (V1, V2, V3, V4)
> > >
> > > - The results show that in **easy problems**, `cube_active` consistently achieves **the best performance**.
> > > - According to **Theorem 1 and Assumption 5**, if the uncertainty is **accurately predicted**, `cube_active` achieves substantial variance reduction, leading to improved performance.
> > > - In easy problems, the uncertainty prediction **tends to be more accurate**, resulting in better performance.
> > > - Nevertheless, as mentioned in **Remark 2**, the `cube_active` method **retains its variance reduction benefits** even when the uncertainty prediction is not perfectly accurate in practice.
> > > - The results of the hard problems show that `cube_active` still outperforms the baseline methods.
> > > - The computational cost remains **manageable**: in our implementation, the cube method has a complexity of $\mathcal{O}(N)$ (Tillé, 2011), which is unaffected by the difficulty of problems
> > >
> > >
> > >
> > > **Table 2.1: Bike Sharing, RMSE**
> > >
> > > | Problem | `cube_active` | `poisson_active` | `uniform` | `classical` |
> > > | ------- | ------------- | ---------------- | --------- | ----------- |
> > > | Easy    | **0.130284**  | 1.776984         | 2.202898  | 5.792181    |
> > > | Hard    | 1.121787      | 2.084508         | 2.421806  | 5.792181    |
> > >
> > > **Table 2.2: Credit Fraud Detection, RMSE**
> > >
> > > | Problem | `cube_active` | `poisson_active` | `uniform` | `classical` |
> > > | ------- | ------------- | ---------------- | --------- | ----------- |
> > > | Easy    | **0.000144**  | 0.000205         | 0.000220  | 0.000331    |
> > > | Hard    | 0.000228      | 0.000266         | 0.000293  | 0.000331    |
> > >
> > > > ## Inference vs. Learning
> > >
> > > - For **model learning** (e.g., training), it may be preferable to employ methods specifically optimized for predictive performance.
> > > - For **inference tasks** (e.g., population parameter estimation), our proposed framework offers advantages rooted in statistical rigor and theoretical guarantees.
> > >
> > >
> > > Once again, we sincerely appreciate your recognition of our work. We hope these responses adequately address your questions.
> > >
> > > ## Reference
> > >
> > > - Tillé, Y. (2011). Ten years of balanced sampling with the cube method: an appraisal. Survey methodology, 37(2), 215-226.

---

### Official Review · Reviewer_WQHA · 2025-07-03

**Clarity:** 3
**Significance:** 2
**Originality:** 2
**Rating:** 4
**Confidence:** 4

**Summary:**

This paper introduces balanced active inference, a method for estimating the label mean when only a small portion of a large unlabelled data pool can be labeld. The proposed method first learns uncertainty scores and then draws balanced samples with the cube method. After the selected points are labelled, they are used to produce the estimated mean. This paper proves asymptotic normality and shows that the proposed estimator’s variance is lower than that of the baselines. Experiments on synthetic datasets demonstrate better mean estimates comparing to previous methods.

**Questions:**

How does Theorem 1 reflect on the choice of $\\pi$ other than simply $\\mathbb{E}[\\pi_1]$? For example, does Theorem 1 hold for any form of the probability $\\pi$ satisfying the balanced sampling constraint? Clarifying this would help the readers to understand how tightly the theoretical benefit is coupled to the particular $\\pi$ defined in the paper.

**Ethical Concerns:**

["NO or VERY MINOR ethics concerns only"]

**Final Justification:**

As mentioned by the authors, the concerns over the theoretical results are not easy to address. However, the additional experiments make the paper more complete, and thus I have raised my rating.

**Limitations:**

yes

**Quality:**

3

**Strengths And Weaknesses:**

**Strengths**

The problem formulation is clearly formulated, the method and results are cleanly presented. The proposed method can be easily implemented, and the experiments support the theoretical claim. In general, the proposed method is conceptually elegant and has the potential to be practically impactful.

**Weaknesses**

* The scope the paper is very narrow, as it only focuses on one-dimensional mean estimations. Although the presentation is clean within the scope, the narrow scope reduces the significance for general ML audience. For example, can the proposed method, both theory and algorithm, be applied to classification tasks?
* The only theoretical result, Theorem 1, is essentially an application of standard tools, e.g., Chebyshev’s inequality and central limit theorem, on top of existing work (Lemma 1 to 3).  A novel proof technique, or results beyond asymptotic normality, would strengthen the paper given the narrow scope.
* Related to the above point, the Assumption 5 is very strong. It basically asks for an accurate uncertainty estimator. Is obtaining an accurate uncertainty estimator easier than a similarly accurate label estimator itself?
* It is mentioned in the paper that the mixing hyperparameter $\\tau$ is chosen to be 0.5. It would be interesting to see how this hyperparameter influence the performance both theoretically and empirically.
* Some of the claims in the paper are not well supported. For example, it is said "safeguard against distributional shifts," but there seems to be no evidence to support this.

---

> ### Author Rebuttal · Authors · 2025-07-29
>
> # **Response to Reviewer WQHA**
>
> We sincerely appreciate your recognition of the novel methodological contribution, compelling experimental validation, and significant practical implications of our work, as well as your valuable feedback and critiques, which are crucial for refining our paper. Below we provide point-by-point responses that incorporate your suggestions.
>
> ## **Responding to Weaknesses**
>
> > ### **Weakness 1. Scope of the Paper**
>
> Thanks for raising concerns about the scope of the paper. While the generation to M-estimation was briefly discussed in the main text, we will delve deeper by formally extending the Balanced Active Inference framework to the general M-estimation setting, **covering both regression and classification**, and present some experimental results.
>
> #### **(i) Settings of M-estimation**
>
> * Given a family of functions $f(X_i;\theta)$, the task is to solve $ \theta^{\ast}= \arg\min_{\theta} \mathbb{E} \left [L\bigl(X_1, Y_1; \theta\bigr)\right]$.
> * $L(X_1, Y_1; \theta)$ measures the discrepancy between the true label $Y_1$ and the prediction $f(X_1;\theta)$.
> * Our goal is to estimate $\theta^{\ast}$ using a small labeled set $ \mathcal{D}_l$ and a larger unlabeled set $ \mathcal{D}_u $.
> * $ f(X_i;\hat{\theta}) $ is an initial estimator trained on $ \mathcal{D}_l $.
>
> #### **(ii) Uncertainty Measures**
>
> - **Regression.** We use the absolute residual $ |Y_i - f(X_i;\hat{\theta})| $ as the uncertainty measure.
> - **Classification.** Let $ p(X_i) = (p_1(X_i),\ldots,p_K(X_i))$ denote the predicted class probabilities. We define $$u(X_i)=\frac{K}{K-1}\Bigl(1-\max_{j\in[K]} p_j(X_i)\Bigr),$$
> which peaks when the model is maximally uncertain (uniform distribution) and drops to zero when the model is highly confident in one class.
>
> #### **(iii) M-Estimation**
>
> * A model $\hat{u}(\cdot)$ is trained on $\mathcal{D}_l$ to predict the uncertainty of points in  $\mathcal{D}_u $.
>
> * Given a labeling budget $n_b$, adopt the sampling scheme described in Equation (2) of the paper.
>
> * Generate an assignment via the cube method, ensuring the balancing constraint (Equation (4)) is satisfied.
>
> * The proposed estimator is
>
>   $$\tilde{\theta}=\underset{\theta}{\arg \min } \frac{1}{n} \sum_{i=1}^n\{L(X_i, f(X_i ; \hat{\theta}) ; \theta)+[L(X_i, Y_i ; \theta)-L(X_i, f(X_i ; \hat{\theta}) ; \theta)] \frac{\xi_i}{\\pi(X_i)}\},$$
>
>   where $\pi(X_i)$ is the inclusion probability.
>
> #### **(iv) Experiment Results**
>
> * Experiments are conducted on both synthetic data (Table W1.1) and the Bike Sharing dateset (Table W1.2).
>
> **Table W1.1 Synthetic Linear, RMSE**
>
> | Budget | Sample Size | `cube_active` | `poisson_active` | `uniform` | `classical` |
> | ------ | ----------- | ------------- | ---------------- | --------- | ----------- |
> | 0.1    | 500         | **0.0193**    | 0.0300           | 0.0480    | 0.0464      |
> | 0.2    | 1000        | **0.0128**    | 0.0183           | 0.0309    | 0.0325      |
>
> **Table W1.2 Bike Sharing, RMSE (First Feature: `temp`)**
>
> | Budget | Sample Size | `cube_active` | `poisson_active` | `uniform` | `classical` |
> | ------ | ----------- | ------------- | ---------------- | --------- | ----------- |
> | 0.1    | 869         | **32.014**    | 44.279           | 107.979   | 101.781     |
> | 0.2    | 1738        | **21.340**    | 29.348           | 69.920    | 67.390      |
>
> * The results show that our balanced active inference approach has the best performance in terms of RMSE.
>
> A section on M-estimation will be added in the revised version.
>
> > ### **Weakness 2. Limited novelty in theoretical contribution**
>
> - The theoretical foundations of the cube method are still under development.
> - As noted in the **most recent work** by Davezies et al. (2024), even in simplified settings such as standard survey sampling without model uncertainty, only aysmptotic results are avaliable.
> - Non-asymptotic guarantees remain elusive due to **(1) the complex dependence induced by the balancing step**, and **(2) the lack of general concentration tools for such designs**.
> - Given these challenges, our asymptotic analysis represents a meaningful and necessary step toward understanding the statistical properties of balanced active learning.
> - We view non-asymptotic extensions as an important and ambitious direction for future research, but one that lies beyond the scope of this paper.
>
> > ### **Weakness 3. Strength of Assumption 5**
>
> - We acknowledge that the assumption effectively requires an accurate uncertainty estimator, which may not always hold in practice.
>
> - As noted in **Remark 2** of our manuscript, Assumption 5 is primarily introduced to facilitate theoretical analysis, **particularly to derive an explicit form of the asymptotic variance**.
>
> - This assumption is **not** a strict requirement for the validity of our method.
>
> - Empirical results consistently show that **the proposed estimator achieves substantial variance reduction even when the uncertainty estimators are imperfect**, suggesting robustness to mild violations of this assumption.
>
> > ### **Weakness 4. Sensitivity Analysis on $\tau$**
>
> Thank you for this important suggestion. We acknowledge that the choice of $\tau = 0.5$ warrants explicit justification.
>
> - Sensitivity analysis on $\tau$ across multiple datasets will be added in the revised version.
> - The setup is: fix the sampling budget at 0.2 and compute the RMSE and interval width (analogous to Figure 1 in the main text, but with $\tau$ ranging from 0 to 1 on the x-axis) for methods.
> - Below is a representative subset of our experimental results:
>
> **Table W4.1 Bike Sharing, RMSE**
>
> | $\\tau$ | `cube_active` | `poisson_active` | `uniform` | `lassical` |
> | ------- | ----------- | -------------- | ------- | --------- |
> | 0.00    | $\quad$ **0.127**   | $\quad$1.460          | 1.466   | 3.846     |
> | 0.25    | $\quad$**0.093**   | $\quad$1.261          | 1.466   | 3.846     |
> | 0.50    | $\quad$**0.079**   | $\quad$1.146          | 1.466   | 3.846     |
> | 0.75    | $\quad$**0.072**   | $\quad$1.064          | 1.466   | 3.846     |
> | 1.00    | $\quad$**0.130**   | $\quad$1.038          | 1.466   | 3.846     |
>
> **Table W4.2 Bike Sharing,  Interval Width**
>
> | $\\tau$ | `cube_active` | `poisson_active` | `uniform` | `lassical` |
> | ------- | ----------- | -------------- | ------- | --------- |
> | 0.00    | $\quad$ **0.395**   | $\quad$ 4.847          | 4.844   | 12.805    |
> | 0.25    | $\quad$ **0.302**   | $\quad$ 4.128          | 4.844   | 12.805    |
> | 0.50    | $\quad$ **0.254**   | $\quad$ 3.754          | 4.844   | 12.805    |
> | 0.75    | $\quad$ **0.230**   | $\quad$ 3.530          | 4.844   | 12.805    |
> | 1.00    | $\quad$ **0.275**   | $\quad$ 3.432          | 4.844   | 12.805    |
>
> **Table W4.3 Friedman, RMSE**
>
> | $\\tau$ | `cube_active` | `poisson_active` | `uniform` | `lassical` |
> | ------- | ----------- | -------------- | ------- | --------- |
> | 0.00    | $\quad$ **0.029**   | $\quad$ 0.069          | 0.068   | 0.137     |
> | 0.25    | $\quad$ **0.029**   | $\quad$ 0.063          | 0.068   | 0.137     |
> | 0.50    | $\quad$ **0.031**   | $\quad$ 0.060          | 0.068   | 0.137     |
> | 0.75    | $\quad$ **0.034**   | $\quad$ 0.060          | 0.068   | 0.137     |
> | 1.00    | $\quad$ **0.063**   | $\quad$ 0.079          | 0.068   | 0.137     |
>
> **Table W4.4 Post-election survey research, RMSE**
>
> | $\\tau$ | `cube_active` | `poisson_active` | `uniform` | `lassical` |
> | ------- | ----------- | -------------- | ----------- | --------- |
> | 0.00    | $\quad$ **0.00841** | $\quad$ 0.00865        | 0.00862     | 0.01165   |
> | 0.25    | $\quad$ **0.00806** | $\quad$ 0.00840        | 0.00862     | 0.01165   |
> | 0.50    | $\quad$ **0.00790** | $\quad$ 0.00824        | 0.00862     | 0.01165   |
> | 0.75    | $\quad$ **0.00799** | $\quad$ 0.00839        | 0.00862     | 0.01165   |
> | 1.00    | $\quad$ 0.01014     | $\quad$ 0.01047        | **0.00862** | 0.01165   |
>
> - All coverage rates in the above results closely approximate the target confidence level, confirming the validity of our findings.
> - Across all datasets, the RMSE and interval width **generally maintained the following ordering**: `cube_active < poisson_active < uniform < classical`.
> - At $\tau = 0$: The performance of `poisson_active` becomes nearly equivalent to `uniform`, as both rely solely on simple random sampling.
> - As $\tau$ increases, both `cube_active` and `poisson_active` exhibit **precision improvement** (decreasing RMSE and interval width), **reaching lower value** around $\tau = 0.5$ for different datasets. This demonstrably quantifies the precision gains enabled by active sampling.
> - When near $\tau = 1$, RMSE and interval width increase on some datasets. This occurs because the model for estimating prediction uncertainty ($\hat{u}$) is inherently imperfect. When the model erroneously assigns near-zero uncertainty ($\hat{u} \approx 0$) to specific data points, it can significantly inflate the estimator variance. These findings are well-aligned with conclusions established in Zrnic & Candès (2024).
>
> > ### **Weakness 5. Claims not well supported**
>
> Thanks for pointing out this issue. We will carefully check our claims, and add sufficient support or delete non-supported ones.
>
> ## **Responding to Question**
>
> > ### **Q1. $\pi$ in Theorem 1**
>
> Thank you for the insightful question.
>
> - Theorem 1 holds for any $\\{\pi_i\\}_{i=1}^n$ as long as Assumption 3 in the paper is satisfied.
> - The balanced sampling constraint (Equation (4)) is enforced by the Cube method after a $\\{\pi_i\\}_{i=1}^n$ is given.
> - Therefore, the variance reduction benefits proven in Theorem 1 are not tightly coupled to a specific construction of $\\{\pi_i\\}_{i=1}^n$.
> - We will clarify this point in the revision to avoid potential confusion.
>
> ## References
>
> - Davezies, L., Hollard, G., & Merino, P. V. (2024). Revisiting randomization with the cube method. *arXiv:2407.13613*.
> - Zrnic, T., & Candès, E. J. (2024, July). Active statistical inference. *ICML*.

---

> > ### Comment · Reviewer_WQHA · 2025-08-04
> >
> > I thank the author for the detailed response. As mentioned by the authors, the concerns over the theoretical results are not easy to address. However, the additional experiments make the paper more complete, and thus I have raised my rating.

---

> > > ### Author Response · Authors · 2025-08-06
> > >
> > > We highly appreciate your comments and the increased rating.
> > >
> > > In the revision, we will
> > > - Move the brief introduction of **Balanced Active Inference** under the **M-estimation** framework from the discussion section into the main body of the paper, and provide a more detailed formulation covering both regression and classification tasks;
> > > - Augment the experiments section with **additional results under the M-estimation setting** to further support the proposed approach.
> > >
> > > Thank you again for your valuable feedback and support.

---

### Official Review · Reviewer_f7YV · 2025-07-17

**Clarity:** 3
**Significance:** 3
**Originality:** 3
**Rating:** 4
**Confidence:** 4

**Summary:**

This paper addresses the statistical inefficiency caused by traditional independent sampling methods. While existing active inference approaches improve upon uniform sampling by prioritizing instances with high model uncertainty, their reliance on independent selection can lead to imbalanced labeled sets and inflated variance.

To solve this, the authors propose Balanced Active Inference, an algorithm that integrates balanced sampling principles into the active learning process. The core idea is to treat the model's uncertainty estimates as auxiliary variables and enforce a balancing constraint during label selection.

**Questions:**

I have a few concerns on the tractability of this method. Specifically,

- Could you provide some insight into the computational complexity of the cube method as used in your algorithm? How does its runtime scale with the size of the unlabeled pool n? Are there potential scalability challenges when applying this method to massive datasets with millions of unlabeled instances?

- A simpler approach to achieve some form of balance would be to stratify the data based on quantiles of the uncertainty and then sample proportionally from each stratum. How would you compare your method, which achieves exact continuous balancing, to such a stratified sampling approach, both theoretically and empirically?

**Ethical Concerns:**

["NO or VERY MINOR ethics concerns only"]

**Final Justification:**

My concerns are addressed and I shall maintain the score.

**Limitations:**

I have mentioned the limitations I foresee in questions and weaknesses sections above.

**Quality:**

3

**Strengths And Weaknesses:**

*Strengths*

- The idea of using model uncertainty as the balancing variable within the cube method framework is interesting. It directly addresses a well-known weakness of uncertainty-based sampling (variance inflation) with a principled statistical solution.

- The paper is exceptionally well-written. The authors do a good job of motivating the problem, explaining the limitations of prior work, and introducing their proposed solution.

- The experimental validation is thorough and convincing. The authors test their method on a wide array of synthetic and real-world datasets, covering both regression and classification tasks. The use of multiple metrics (RMSE, confidence interval width, and coverage rate) provides a complete picture of the method's performance.


*Weaknesses*

- The paper leverages the cube method, which may be unfamiliar to parts of the core machine learning audience. While it is a standard technique in another field, a slightly more intuitive explanation of its mechanics in the main text could improve accessibility. Furthermore, the computational complexity of the cube method itself, especially for very large unlabeled datasets, is not discussed, which could be a practical concern for scalability.

---

> ### Author Rebuttal · Authors · 2025-07-29
>
> # **Response to Reviewer f7YV**
>
> Thank you for your recognition of the novelty and methodological rigor of our research, as well as the thorough experimental validation presented. We greatly appreciate your constructive feedback and critiques, which are invaluable for refining our manuscript. Below are point-by-point responses incorporating your suggestions:
>
> ## **Responding to Weaknesses**
>
> > ### **Weakness 1. Intuition of Cube Method**
>
> Thanks for the valuable suggestion of providing **a more intuitive explanation of the cube method**. We will add the following explanation in the revised version:
>
> - Covariate balancing is a commonly used statistical technique to **improve statistical efficiency** when designing a sampling scheme.
>
> - The cube method (Deville & Tillé, 2004) achieves covariate balancing by consecutively updating selection probabilities to satisfy certain balancing constraints.
>
> - Theoretically, the cube method achieves covariate balancing by sacrificing independence among sampled elements, making it challenging to investigate its asymptotic properties.
>
> - Practically, the cube method is easy to implement and can be seamlessly integrated in active inference.
>
> ## **Responding to Questions**
>
> > ### **Q1. Computational Complexity of Cube Method**
>
> - As noted in Tillé (2011), the computational complexity of the cube method is $\mathcal{O}(N \times p^2)$, where $N$ is the **population size** and $p$ the **number of balancing covariates**.
>
> - In our implementation, **only one** auxiliary covariate ($\hat{u}$, the uncertainty measure) is used ($p=1$). Thus, complexity reduces to $\mathcal{O}(N)$, meaning runtime scales linearly with the unlabeled size $n$.
>
> - This linear scaling enables efficient handling of large datasets.
>
> - A discussion on the computational complexity of the cube method will be added in the main text.
>
> > ### **Q2. Stratified Sampling**
> - Thanks for the thoughtful suggestion of trying a stratified sampling approach.
> - New experiments are implemented including the **baseline** of **stratified sampling**: stratifying by $\hat{f}(X)$ (predicted labels; `Stratified_Y`) and $\hat{u}$ (predicted uncertainty; `Stratified_u`) using 5 quantile bins (width=0.2) and sampling proportionally from each stratum.
>
> - Below is a representative subset of our experimental results:
>
> **Table 1 Bike Sharing, RMSE**
>
> | Budget | Sample Size | `Poisson_Active` | `Cube_Active` | `Stratified_Y` | `Stratified_u` |
> |--------|-------------|----------------|-------------|--------------|--------------|
> | 0.1    | 869         | 1.777          | **0.128**       | 1.754        | 1.795        |
> | 0.2    | 1738        | 1.146          | **0.079**       | 1.141        | 1.140        |
>
> **Table 2 Friedman, RMSE**
>
> | Budget | Sample Size | `Poisson_Active` | `Cube_Active` | `Stratified_Y` | `Stratified_u` |
> |--------|-------------|----------------|-------------|--------------|--------------|
> | 0.1    | 500 | 0.093    | **0.046**       | 0.093        | 0.092|
> | 0.2    | 1000        | 0.060     | **0.030**       | 0.061        | 0.060|
>
> **Table 3 Credit Fraud Detection, RMSE**
>
> | Budget | Sample Size | `Poisson_Active` | `Cube_Active` | `Stratified_Y` | `Stratified_u` |
> |--------|-------------|----------------|-------------|--------------|--------------|
> | 0.15   | 21,360         | 0.000155     | **0.000120**       | 0.000160        | 0.000150|
> | 0.35   | 49,841      | 0.000088     | **0.000066**       | 0.000090        | 0.000087|
>
> - Experiments on the datasets show that cube sampling achieves significantly lower RMSE due to its exact continuous balancing.
>
> - Stratification balances auxiliary covariates **discretely**, whereas the cube method enforces exact population-level balance for **continuous** $\hat{u}$. This eliminates bias in uncertainty estimation, which is critical for active inference performance.
>
> ## References
> - Deville, J. C., & Tillé, Y. (2004). Efficient balanced sampling: the cube method. Biometrika, 91(4), 893-912.
> - Tillé, Y. (2011). Ten years of balanced sampling with the cube method: an appraisal. Survey methodology, 37(2), 215-226.

---

> ### Author Response · Authors · 2025-08-07
>
> We highly appreciate your thoughtful review and constructive feedback. Based on your comments, we have incorporated the following revisions:
>
> ## **Revisions addressing your specific comments:**
>
> - Provide an **intuitive and comprehensive** explanation of the **Cube Method.**
> - Clarify the $\mathcal{O}(N)$ **computational complexity** of balanced sampling.
> - Include **stratified sampling** as an additional baseline in our experiments.
>
> ## **Further enhancements to strengthen the paper:**
>
> In discussions with other reviewers, we have incorporated the following enhancements into the revised manuscript:
>
> - Present a detailed formulation of our approach within the **M-estimation** framework, covering both regression and classification tasks.
> - Provide **additional experimental results** under the M-estimation framework, demonstrating **consistent improvements** over baselines.
> - Add more **comparative experiments** including:
>     1. **Sensitivity analysis** of key parameters;
>     2. Performance comparison on **easy vs. hard problems**;
>     3. **Coverage rate analysis** under different confidence levels.
>
> If you have any further suggestions or questions, we would be grateful for the opportunity to address them. Once again, thank you for your thoughtful review and valuable support.

---

> ### Comment · Area_Chair_CUH8 · 2025-08-08
>
> Dear Reviewer f7YV,
>
> Please note that the final justification is invisible to the authors and will only be revealed to them after the decision. Could you please leave them a comment to let them know whether you are satisfied with their response?
>
> Best regards,
>
> AC

---

### Comment · Area_Chair_CUH8 · 2025-08-05
**Reminder to Reviewers to engage in discussion with Authors**

Dear Reviewers,

Please read the authors' rebuttals if you have done so yet, and respond to them as soon as possible to allow sufficient time for follow-up exchanges. The Author-Reviewer discussion is crucial to a constructive reviewing process, to which your reactivity and engagement are indispensable.

Best regards,

AC

---

### Note · Authors · 2025-08-13

We appreciate the reviewers for their constructive feedback and the AC for diligent oversight. The recognition of our work’s novelty and theoretical analysis is highly encouraging.

Below, we summarize the **key contributions** of our work and the **corresponding revisions** made in response to the reviewers’ comments and discussions.

## **Contributions:**

- **Innovatively increase** the statistical efficiency of **active inference** through **balanced sampling**
- Incorporate model **uncertainty** as auxiliary covariate in balanced sampling using a **cube method**
- Provide **closed-form expressions** for the asymptotic variance, illustrating the **variance reduction property** of the proposed method
- Demonstrate the **broad applicability and superiority** of our approach through extensive experiments on diverse real-world and synthetic datasets

## **Revisions**:

**1. Introduction**
-   Highlight the **innovations** and **distinct contributions** of our approach from existing methods
-   Clearly articulate the problem our paper aims to address, namely reducing sampling-induced variance by balanced sampling
-   Systematically introduce the contributions and limitations of existing Active Learning/Inference research

**2. Problem Setup & Balanced Active Inference**
-   Provide a **detailed overview** of existing Active Learning/Inference methodologies
-   Introduce our method under the **M-estimation** framework and provide a detailed formulation covering both **regression** and **classification** tasks
-   Clarify the $\mathcal{O}(N)$ computational **complexity** of balanced sampling
-   Clearly present our methodological improvements and their motivations
-   Provide an intuitive and comprehensive explanation of the cube method

**3. Theoretical Properties**
-   Offer more **intuitive interpretations** of the remarks and theorems (e.g., Remark 1)
-   Provide a detailed interpretation of our theoretical results


**4. Experiments**
-   Present additional experimental results under the **M-estimation** framework over baselines
-   Conduct **sensitivity analysis** of key parameters
-   Analyze **coverage rates** under varying confidence levels
-   Include **stratified sampling** as an additional baseline
-   Compare performance on easy vs. hard problems

Once again, we appreciate the reviewers for their insightful comments and the AC for the thoughtful coordination, which have greatly helped improve the clarity and quality of our manuscript.

---

### Decision · Program_Chairs · 2025-09-17

**Decision:**

Accept (poster)

**Comment:**

This article looks at a less studied problem, namely active inference, and proposes to improve upon the standard approach of independent sampling by using the cube method to impose a balancing constraint on queried examples. The proposed algorithm is shown to have a smaller asymptotic variance than independent sampling under the assumption of a nearly accurate estimator of uncertainty. The robustness of the variance reduction benefit to this assumption is demonstrated through experiments on synthetic and real data.

The novelty and the interest of the proposed method were appreciated by the reviewers. It is simple, conceptually elegant, and potentially impactful on the field of active inference. Questions on computational complexity, choice of hyperparameter, generalization to M-estimator framework, and comparison to more baselines were raised and addressed during rebuttal, with the support of supplementary empirical results. One remaining concern is about the presentation. The problematics of active inference should be more carefully introduced, to allow an easier grasp of the motivation and the intuition behind the proposed balanced sampling approach to non-expert readers. A discussion on the relation and the difference with active learning would also help better contextualize the contribution of this work.